# Normalized/Clipped SGD with Perturbation for Differentially Private Non-Convex Optimization

## Abstract

By ensuring differential privacy in the learning algorithms, one can rigorously mitigate the risk of large models memorizing sensitive training data. In this paper, we study two algorithms for this purpose, i.e., DP-SGD and DP-NSGD, which first clip or normalize *per-sample* gradients to bound the sensitivity and then add noise to obfuscate the exact information. We analyze the convergence behavior of these two algorithms in the non-convex empirical risk minimization setting with two common assumptions and achieve a rate $\mathcal{O}\left(\sqrt[4]{\frac{d\log(1/\delta)}{N^2\epsilon^2}}\right)$ of the gradient norm for a $d$-dimensional model, $N$ samples and $(\epsilon, \delta)$-DP, which improves over previous bounds under much weaker assumptions. Specifically, we introduce a regularizing factor in DP-NSGD and show that it is crucial in the convergence proof and subtly controls the bias and noise trade-off. Our proof deliberately handles the per-sample gradient clipping and normalization that are specified for the private setting. Empirically, we demonstrate that these two algorithms achieve similar best accuracy while DP-NSGD is comparatively easier to tune than DP-SGD.

## 1 Introduction

Modern applications of machine learning strongly rely on training models with sensitive datasets, including medical records, real-life locations, browsing histories and so on. These successful applications raise an unavoidable risk of privacy leakage, especially when large models are shown to be able to memorize training data (Carlini et al., 2020). Differential Privacy (DP) is a powerful and flexible framework (Dwork et al., 2006b) to quantify the influence of each individual and reduce the privacy risk. Specifically, we study the machine learning problem in the formalism of minimizing *empirical risk* privately:

$$\min_{\boldsymbol{x}\in\mathbb{R}^d} f(\boldsymbol{x}) \triangleq \mathbb{E}_\xi[\ell(\boldsymbol{x}, \xi)] = \frac{1}{N}\sum_{i=1}^{N}\ell(\boldsymbol{x}, \xi_i), \tag{1}$$

where the objective $f(\boldsymbol{x})$ is an empirical average of losses evaluated at each data point $\xi$ and $\xi$ is sampled uniformly from the given dataset $\{\xi_i, 1 \leq i \leq N\}$[①].

In order to provably achieve the privacy guarantee, one popular algorithm is *differentially private stochastic gradient descent* or DP-SGD for abbreviation, which clips per-sample gradients with a preset threshold and perturbs the gradients with Gaussian noise at each iteration. Formally, given a set of gradients $\{\boldsymbol{g}^{(i)}, i \in \mathcal{S} \subset [N]\}$ where $\mathcal{S}$ can be thought of as a set of indices of gradients in a mini-batch and $\boldsymbol{g}^{(i)} = \nabla_{\boldsymbol{x}}\ell(\boldsymbol{x}, \xi_i)$ is the gradient computed with some data point $i$, a threshold $c > 0$, a learning rate $\eta > 0$ and a noise multiplier $\sigma$, the updating rule of DP-SGD goes from $\boldsymbol{x}$ to the following

$$\boldsymbol{x}^+ = \boldsymbol{x} - \eta\left(\frac{1}{|\mathcal{S}|}\sum_{i\in\mathcal{S}}\bar{h}^{(i)}\boldsymbol{g}^{(i)} + \bar{\boldsymbol{z}}\right), \tag{2}$$

---

[①]Our method later uses uniform sub-sampling without replacement to construct mini-batches. Thus, the second equality in 1 is exact.

where $\bar{z} \sim \mathcal{N}(0, c^2\sigma^2 \boldsymbol{I}_d)$ is an isotropic Gaussian noise and $\bar{h}^{(i)} = \min\{1, c/\|\boldsymbol{g}^{(i)}\|\}$ is the per-sample clipping factor. Intuitively speaking, the per-sample clipping procedure controls the influence of one individual. DP-SGD (Abadi et al., 2016) has made a benchmark impact in deep learning with differential privacy, which is also referred to as the *gradient perturbation* approach. Despite being applied into many fields (Hassan et al., 2019; Ji et al., 2014), it has also been extensively studied from many aspects, e.g., convergence (Bassily et al., 2014; Yu et al., 2020), privacy analysis (Abadi et al., 2016), adaptive clipping threshold (Asi et al., 2021; Andrew et al., 2019; Pichapati et al., 2019), hyperparameter choices (Li et al., 2021; Papernot et al., 2020a; Papernot & Steinke, 2021; Mohapatra et al., 2021) and so forth (Bu et al., 2020; 2021; Papernot et al., 2020b).

Another natural option to achieve differential privacy is *normalized gradient with perturbation*, which we coin "DP-NSGD". It normalizes per-sample gradients to control individual contribution and then adds noise accordingly. The update formula is the same as (2) except replacing $\bar{h}^{(i)}$ with a *per-sample normalization factor*

$$h^{(i)} = \frac{1}{r + \|\boldsymbol{g}^{(i)}\|}, \tag{3}$$

and replacing $\bar{z}$ with $\boldsymbol{z} \in \mathcal{N}(0, \sigma^2 \mathbf{I}_d)$ since each sample's influence is normalized to be 1. In (3), we introduce a regularizer $r > 0$, which not only addresses the issue of ill-conditioned division but also controls the bias and noise trade-off as we will see in the analysis. A concurrent work of ours Bu et al. (2023) also proposed exactly the same method, but their theoretical results are based on different assumptions, while the experimental findings are mutually supportive.

An intuitive thought that favor DP-NSGD is that the clipping threshold is hard to choose due to the changing statistics of the gradients over the training trajectory (Andrew et al., 2019; Pichapati et al., 2019). In more details, the injected Gaussian noise $\eta\bar{z}$ in (2) is proportional to the clipping threshold $c$ and this noise component would dominate over the gradient component $\sum \bar{h}^{(i)} \boldsymbol{g}^{(i)}/|\mathcal{S}|$, when the gradients $\|\boldsymbol{g}^{(i)}\| \ll c$ are getting small as optimization algorithm iterates, thus hindering the overall convergence. DP-NSGD aims to alleviate this problem by replacing $\bar{h}^{(i)}$ in (2) with a *per-sample gradient normalization factor* $h^{(i)}$ in (3), thus enhancing the signal component $g^{(i)}$ when it is small.

It is obvious that both clipping and normalization introduce *bias*[2] that might prevent the optimizers from converging (Chen et al., 2020; Zhao et al., 2021). Most of previous works on the convergence of DP-SGD (Bassily et al., 2014; Asi et al., 2021; Yu et al., 2020) neglect the effect of such biases by assuming a global gradient upper bound of the problem, which does not exist for the cases of deep neural network models. Chen et al. (2020) have made a first attempt to understand gradient clipping, but their results strongly rely on a symmetric assumption which is not that realistic.

In this paper, we consider both the effect of per-sample normalization/clipping and the injected Gaussian perturbation in the convergence analysis. If properly setting the hyperparameters, we achieve an $\mathcal{O}\left(\sqrt[4]{\frac{d \log(1/\delta)}{N^2 \epsilon^2}}\right)$ convergence rate of the gradient norm for the general non-convex objective with a $d$-dimensional model, $N$ samples and $(\epsilon, \delta)$-DP, under only two weak assumptions $(L_0, L_1)$-generalized smoothness (Zhang et al., 2020b) and $(\tau_0, \tau_1)$-bounded gradient variance. These assumptions are very mild as they allow the smoothness coefficient and the gradient variance growing with the norm of gradient, which is widely observed in the setting of deep learning.

Our contributions are summarized as follows.

- For the *differentially private empirical risk minimization*, we establish the convergence rate of the DP-NSGD and the DP-SGD algorithms for general non-convex objectives with $(L_0, L_1)$-smoothness condition and $(\tau_0, \tau_1)$-bounded gradient variance, and explicitly characterizes the bias of the per-sample clipping or normalization. In particular, our utility bounds match the best convergence rates that are available, even under weakened conditions.

---

[2]Here *bias* means that the expected descent direction differs from the true gradient $\nabla f$.

Table 1: Expected gradient norm bounds (the smaller, the better) for non-convex empirical risk minimization with/without $(\epsilon, \delta)$-DP guarantee. All the algorithms assume certain bound on gradient noises, which may be different from one another. Notations: $N, T$ and $d$ are the number of samples, iterations and parameters, respectively. All bounds should be read as $\mathcal{O}(\cdot)$ and $\log \frac{1}{\delta}$ is omitted.

| Algorithm | Smoothness | Condition on gradient estimate $\boldsymbol{g}$ | Bias handled in analysis$^\diamond$ | Gradient norm bound |
|---|---|---|---|---|
| SGD
Ghadimi & Lan (2013) | $L$ | $\mathbb{E}\|\boldsymbol{g} - \nabla f\|^2 \leq V$ | N/A | $\mathcal{O}\left(\frac{1}{\sqrt{T}} + \frac{\sqrt{V}}{\sqrt[4]{T}}\right)$ |
| Clipped SGD
Zhang et al. (2020b) | $(L_0, L_1)$ | $\mathbb{E}\|\boldsymbol{g} - \nabla f\|^2 \leq V$ | $\checkmark$ | $\mathcal{O}\left(\frac{V^2}{\sqrt{T}} + \frac{V^{3/2}}{\sqrt[4]{T}}\right)$ |
| DP-NormFedAvg
Das et al. (2021) | $L$ | quasar-Convex | $\checkmark$ | $\mathcal{O}\left(\frac{D_X \sqrt{d}}{N\epsilon} + \mathbb{E}_i \|\boldsymbol{x}_i^* - \boldsymbol{x}^*\|\right)^\dagger$ |
| DP-GD
Wang et al. (2019a) | $L$ | $\|\boldsymbol{g}\| \leq \tau$ a.s. | $\times$ | $\mathcal{O}\left(\sqrt[4]{\frac{d}{N^2 \epsilon^2}}\right)$ |
| DP-SRM
Wang et al. (2023) | $L$ | $\|\boldsymbol{g}\| \leq \tau$ a.s. | $\times$ | $\mathcal{O}\left(\sqrt[4]{\frac{d}{N^2 \epsilon^2}}\right)$ |
| DP-GD/RMSprop/Adam
Zhou et al. (2020) | $L$ | $\|\boldsymbol{g}\| \leq \tau$ a.s. | $\times$ | $\mathcal{O}\left(\sqrt[4]{\frac{d}{N^2 \epsilon^2}}\right)$ |
| DP-(N)SGD
Bu et al. (2023) (concurrent) | $L$ | $\mathbb{E}\|\boldsymbol{g} - \nabla f\|^2 \leq V$
$\boldsymbol{g}$ centrally symmetric around its mean | $\checkmark$ | $\mathcal{O}\left(\sqrt[4]{\frac{d}{N^2 \epsilon^2}}\right)$ |
| DP-(N)SGD **(Ours)** | $(L_0, L_1)$ | $\|\boldsymbol{g} - \nabla f\| \leq \tau_0 + \tau_1 \|\nabla f\|$ a.s. | $\checkmark$ | $\mathcal{O}\left(\sqrt[4]{\frac{d}{N^2 \epsilon^2}}\right)^\ddagger$ |

$^\dagger$ More remarks on this line are packed in Appendix A.
$^\ddagger$ To be preciser, Corollary 3.5 suggests DP-SGD achieving exactly this rate, but Corollary 3.3 also adds another non-vanishing term for DP-NSGD.
$^\diamond$ When analyzing the utility theoretically, most previous works in the literature of differentially private optimization do not address the influence of gradient clipping/regularization. Accordingly, they usually assume a much stronger condition that $\|\boldsymbol{g}\| \leq \tau$ a.s.

- For the DP-NSGD algorithm, we introduce a regularizing factor which turns out to be crucial in the convergence analysis and induces interesting trade-off between the bias due to normalization and the decaying rate of the upper bound.

- We identify one key difference in the proofs of DP-NSGD and DP-SGD. As the gradient norm approaches zero, DP-NSGD cannot guarantee the function value to drop along the expected descent direction, and introduce a non-vanishing term that depends on the regularizer and the gradient variance.

- We evaluate the empirical performance of DP-NSGD and DP-SGD respectively on deep models with $(\epsilon, \delta)$-DP and show that they both can achieve comparable accuracy but the former is easier to tune than the later.

The paper is organized as follows. After introducing the problem setup in Section 2, we present the algorithms and theorems in Section 3 and show numerical experiments on vision and language tasks in Section 4. We make concluding remarks in Section 5.

## 2 Problem Setup

### 2.1 Notations

Denote the private dataset as $\mathbb{D} = \{\xi_i, 1 \leq i \leq N\}$. The loss $\ell(\boldsymbol{x}, \xi_i)$ is defined for every model parameter $\boldsymbol{x} \in \mathbb{R}^d$ and data record $\xi_i$. In the sequel, $\|\boldsymbol{x}\|$ is denoted as the $\ell_2$ norm of a vector $\boldsymbol{x} \in \mathbb{R}^d$, without other specifications. From time to time, we interchangeably use $\nabla_{\boldsymbol{x}} \ell(\boldsymbol{x}, \xi_i)$ and $\boldsymbol{g}^{(i)}$ to denote the gradient of $\ell(\cdot, \cdot)$ w.r.t. $\boldsymbol{x}$ evaluated at $(\boldsymbol{x}, \xi_i)$. We are given an oracle to draw a mini-batch $\mathcal{B}$ of data for each iteration. Our target is to minimize the empirical average loss (1) satisfying $(\epsilon, \delta)$-differential privacy .

**Definition 2.1** ($(\epsilon, \delta)$-DP, (Dwork et al., 2006a)). *A randomized mechanism $\mathcal{M}$ guarantees $(\epsilon, \delta)$-differentially privacy if for any two neighboring input datasets $\mathbb{D} \sim \mathbb{D}'$ ($\mathbb{D}'$ differ from $\mathbb{D}$ by substituting one record of data) and for any subset of output $S$ it holds that $\Pr[\mathcal{M}(\mathbb{D}) \in S] \leq e^\epsilon \Pr[\mathcal{M}(\mathbb{D}') \in S] + \delta$.*

Besides, we also define the following notations to illustrate the bound we derived. We write $f(\cdot) = \mathcal{O}(g(\cdot))$, $f(\cdot) = \Omega(g(\cdot))$ to denote $f(\cdot)/g(\cdot)$ is upper or lower bounded by a positive constant. We also write $f(\cdot) = \Theta(g(\cdot))$ to denote that $f(\cdot) = \mathcal{O}(g(\cdot))$ and $f(\cdot) = \Omega(g(\cdot))$. Throughout this paper, we use $\mathbb{E}$ to represent taking expectation over the randomness of optimization procedures: drawing noisy gradients estimates $\boldsymbol{g}$ and adding extra Gaussian perturbation $\boldsymbol{z}$. In the meantime, $\mathbb{E}_k$ takes conditional expectation given $\boldsymbol{x}_k$, the $k$-th iterate of our optimization algorithm.

In the settings of non-convex loss functions $f(\boldsymbol{x})$ in $\boldsymbol{x}$, we measure the utility of some algorithm via bounding the expected minimum gradient norm $\mathbb{E}\left[\min_{0 \le k < T} \|\nabla f(\boldsymbol{x}_k)\|\right]$. If we want to measure utility via bounding function values, an extra convex condition or its weakened versions are essential. See Section 2.3 for more discussion on the last point.

## 2.2 Assumptions on Smoothness and Variance

**Definition 2.2.** We say that a continuously differentiable function $f(\boldsymbol{x})$ is $(L_0, L_1)$-generalized smooth, if for all $\boldsymbol{x}, \boldsymbol{y} \in \mathbb{R}^d$, we have $\|\nabla f(\boldsymbol{x}) - \nabla f(\boldsymbol{y})\| \le (L_0 + L_1 \|\nabla f(\boldsymbol{x})\|)\|\boldsymbol{x} - \boldsymbol{y}\|$.

Firstly appearing in Zhang et al. (2020b), a similar condition is derived via empirical observations that the operator norm $\|\nabla^2 f(\boldsymbol{x})\|$ of the Hessian matrix increases with the gradient norm $\|\nabla f(\boldsymbol{x})\|$ in training language models. If we set $L_1 = 0$, then Definition 2.2 turns into the usually assumed $L$-smoothness. Our first assumption below comprises of this relaxed notion of smoothness, Definition 2.2, and that $f(\boldsymbol{x})$ is lower bounded.

**Assumption 2.1.** *We assume that $f(\boldsymbol{x})$ is $(L_0, L_1)$-generalized smooth, as defined in Definition 2.2. We also set the function value to be lower bounded, $f^* = \inf_{\boldsymbol{x} \in \mathbb{R}^d} f(\boldsymbol{x})$. For notational convenience, write $D_f \triangleq f(\boldsymbol{x}_0) - f^* < \infty$ as the gap in function value between the initialization $\boldsymbol{x}_0$ and the lower bound.*

Many optimization studies necessitate the initialization point $\boldsymbol{x}_0$, to be sufficiently close to the optimal point $\boldsymbol{x}^*$, in terms of Euclidean distance; that is, they require an upper bound on $D_X \triangleq \|\boldsymbol{x}_0 - \boldsymbol{x}^*\|$. However, this assumption may obscure some dependence on the dimension, as $D_X$ tends to scale with the dimension of the model. In our context, such assumptions are not necessary.

Moreover, to handle the stochasticity in gradient estimates, we employ the following *almost sure* upper bound on the gradient variance as another assumption.

**Assumption 2.2.** *For all $\boldsymbol{x} \in \mathbb{R}^d$, $\mathbb{E}[\nabla_{\boldsymbol{x}} \ell(\boldsymbol{x}, \xi)] = \nabla f(\boldsymbol{x})$. Furthermore, there exists $\tau_0 > 0$ and $0 \le \tau_1 < 1$, such that it holds $\|\nabla_{\boldsymbol{x}} \ell(\boldsymbol{x}, \xi) - \nabla f(\boldsymbol{x})\| \le \tau_0 + \tau_1 \|\nabla f(\boldsymbol{x})\|$ almost surely for $\xi$ drawn from the data distribution.*

Since we are focusing on the problem of empirical risk minimization, Assumption 2.2 turns out to be a condition onto the dataset $\mathbb{D} = \{\xi_i, 1 \le i \le N\}$ that there exist constants $(\tau_0, \tau_1)$ uniform in $\boldsymbol{x}$ such that $\|\nabla_{\boldsymbol{x}} \ell(\boldsymbol{x}, \xi_i) - \nabla f(\boldsymbol{x})\| \le \tau_0 + \tau_1 \|\nabla f(\boldsymbol{x})\|$ for all $1 \le i \le N$. We note that similar almost-surely bounds on the gradient noises have been assumed in Wang et al. (2019a; 2023); Zhou et al. (2020). In comparison, Assumption 2.2 is a weakened version: it allows the deviation $\|\nabla_{\boldsymbol{x}} \ell(\boldsymbol{x}, \xi) - \nabla f(\boldsymbol{x})\|$ grows with respect to the gradient norm $\|\nabla f(\boldsymbol{x})\|$, which matches practical observation more closely. This almost-surely type of assumption seems unavoidable for analyzing DP optimization algorithms otherwise the sensitivity of an individual is out of control. One alternative option in literature is adding light-tailed conditions on the distribution of $\nabla_{\boldsymbol{x}} \ell(\boldsymbol{x}, \xi)$ (Fang et al., 2022).

Outside the context of empirical risk minimization, we can also find meaningful examples for which Assumption 2.2 holds and is more reasonable to the concrete setting.

**Example 2.3.** We provide a natural setting in which our Assumption 2.2 provably holds. Consider a linear model $v = \mathbf{w}^T \mathbf{x}^* + u$ and MSE loss $\ell(\mathbf{x}, v, \mathbf{w}) = \frac{1}{2}|v - \mathbf{w}^T \mathbf{x}|^2$, where $v \in \mathbb{R}$ is the response and $\mathbf{w} \in \mathbb{R}^d$ comprises of predictors. Here $u$ denotes mean-zero uncorrelated noise $\mathbb{E}[u\mathbf{w}] = 0, \mathbb{E}[u] = 0$. By normalizing, we assume $|v| \le C_v$ and $\|\mathbf{w}\| \le C_w$ are both bounded, and $\mathbb{E}[\mathbf{w}\mathbf{w}^T]$ has positive minimum eigenvalue $\lambda_{min} > 0$.

In this setup, we find

$$\nabla_{\boldsymbol{x}} \ell(\mathbf{x}, v, \mathbf{w}) = (\mathbf{w}^T \mathbf{x} - v)\mathbf{w} = \mathbf{w}\mathbf{w}^T (\mathbf{x} - \mathbf{x}^*) - u\mathbf{w}$$

and $\nabla L(\mathbf{x}) = \mathbb{E}_{v,\mathbf{w}} \nabla_x \ell(\mathbf{x}, v, \mathbf{w}) = \mathbb{E}[\mathbf{w}\mathbf{w}^T](\mathbf{x} - \mathbf{x}^*)$. Consequently, we will have

$$\|\nabla_{\boldsymbol{x}} \ell(\boldsymbol{x}, v, \boldsymbol{w})\| \leq C_w(C_w \|\mathbf{x}^*\| + C_v) + C_w^2 \|\mathbf{x} - \mathbf{x}^*\| \leq \tau_0 + \tau_1 \|\nabla L(\mathbf{x})\|$$

almost surely for $v, \mathbf{w}$, where $\tau_0 = C_w(C_w\|\mathbf{x}^*\| + C_v)$ and $\tau_1 = C_w^2/\lambda_{min}$.

## 2.3 Related Works

Apart from the literature mentioned in Section 1, there are a large body of works related to our study. We briefly review part of them as follows.

**Private Deep Learning:** Many papers have made attempts to theoretically analyze gradient perturbation approaches in various settings, including (strongly) convex (Chaudhuri & Monteleoni, 2009; Wang et al., 2017; Kuru et al., 2020; Yu et al., 2020; Asi et al., 2021; Kamath et al., 2022; Wang et al., 2022) or non-convex (Wang et al., 2019a; 2022; Zhou et al., 2020; Wang et al., 2023) objectives. However, these papers did not take gradient clipping into consideration, and simply treat DP-SGD as SGD with extra Gaussian noise. Chen et al. (2020) made a first attempt to understand gradient clipping, but their results strongly rely on a symmetric assumption which is considered as unrealistic, which is used in a concurrent work (Bu et al., 2023) of ours to establish the convergence of DP-NSGD.

As for algorithms involving *normalizing*, Das et al. (2021) studied DP-NormFedAvg, a client-level DP optimizer. More detailed remarks are packed up in Appendix A.

These mentioned results are hard to compare due to the differences of the settings, assumptions and algorithms. We present a part of them in Table 1. Full discussions and comparisons are packed up in Appendix A.

**Non-Convex Stochastic Optimization:** Ghadimi & Lan (2013) established the convergence of randomized SGD for non-convex optimization. The objective is assumed to be $L$-smooth and the randomness on gradients is assumed to be light-tailed with factor $V$. We note that the rate $\mathcal{O}(V/\sqrt[4]{T})$ has been shown to be optimal in the worst-case under the same condition (Arjevani et al., 2019).

Outside the privacy community, understanding gradient normalization and clipping is also crucial in analyzing adaptive stochastic optimization methods, including AdaGrad (Duchi et al., 2011), RMSProp (Hinton et al., 2012), Adam (Kingma & Ba, 2014) and normalized SGD (Cutkosky & Mehta, 2020). However, with the average of a mini-batch of gradient estimates being clipped, this *batch* gradient clipping differs greatly from the *per-sample* gradient clipping in the private context. Zhang et al. (2020b) and Zhang et al. (2020a) showed the superiority of batch gradient clipping with and without momentum respectively under $(L_0, L_1)$-smoothness condition for non-convex optimization. Due to a strong connection between clipping and normalization, we also assume this relaxed condition in our analysis. We further explore this condition for some specific cases in great details. Zhang et al. (2020c) studied SGD with gradient clipping under heavy-tailed condition for gradient estimation. Cutkosky & Mehta (2021) found that a fine integration of clipping, normalization and momentum, can overcome heavy-tailed gradient variances via a high-probability bound. Jin et al. (2021) discovered that normalized SGD with momentum is also distributionally robust. Significant advancements have been made in understanding convergence rates for non-convex objectives from the perspective of differentially private Riemannian optimization, as highlighted by Han et al. (2022); Utpala et al. (2022a;b) in their improved differentially private frameworks. Notably, many problems are, in fact, geodesically convex, and these frameworks offer utility bounds on the expected empirical excess risk.

# 3 Normalized/Clipped Stochastic Gradient Descent with Perturbation

In this section, we first present the algorithms DP-NSGD and DP-SGD, and their privacy guarantees. Then we establish their convergences, respectively, with proof sketches. In the end, we analyze the biases of these algorithms and verify them with experiments.

## 3.1 Algorithms and Their Privacy Guarantees

---

**Algorithm 1** Differentially Private Normalized Stochastic Gradient Descent, DP-NSGD

**Input:** initial point $\boldsymbol{x}_0$; number of epochs $T$; default learning rates $\eta_k$; mini batch size $B$; noise multiplier $\sigma$; regularizer $r$.

**for** $k = 0$ **to** $T - 1$ **do**

Draw a mini-batch $\mathcal{S}_k$ of size $B$ and compute individual gradients $\boldsymbol{g}_k^i$ at point $\boldsymbol{x}_k$ where $i \in \mathcal{S}_k$.
For $i \in \mathcal{S}_k$, compute per-sample normalizing factor

$$h_k^{(i)} = \frac{1}{r + \|\boldsymbol{g}_k^{(i)}\|}.$$

Draw $\boldsymbol{z}_k \sim \mathcal{N}(0, \sigma^2 \boldsymbol{I}_d)$ and update the parameters by

$$\boldsymbol{x}_{k+1} = \boldsymbol{x}_k - \eta_k \left( \frac{1}{B} \sum_{i \in \mathcal{S}_k} h_k^{(i)} \boldsymbol{g}_k^{(i)} + \boldsymbol{z}_k \right).$$

**end for**

---

Since no literature formally displays DP-NSGD in a centralized setting, we present it in Algorithm 1. Compared to the usual SGD update, DP-NSGD contains two more steps: per-sample gradient normalization, i.e., multiplying $\boldsymbol{g}_k^{(i)}$ with $h_k^{(i)}$, and noise injection, i.e., adding $\boldsymbol{z}_k$. The normalization well controls each sample's contribution to the update and the noise obfuscates the exact information.

The well-known DP-SGD (Abadi et al., 2016) replaces the normalization with clipping, i.e., replacing $h_k^{(i)}$ with $\bar{h}_k^{(i)} = \min\{1, c/\|\boldsymbol{g}_k\|\}$ and replacing $\boldsymbol{z}_k$ with $\bar{\boldsymbol{z}}_k \sim \mathcal{N}(0, c^2\sigma^2 \boldsymbol{I}_d)$ in Algorithm 1. DP-SGD introduces a new hyper-parameter, the clipping threshold $c$.

To facilitate the common practice in private deep learning, we adopt *uniform sub-sampling without replacement* for both theory and experiments, instead of *Poisson sub-sampling* originally adopted in DP-SGD (Abadi et al., 2016). Due to this difference, the following lemma shares the same expression as Theorem 1 in Abadi et al. (2016), but requires a new proof. Deferred in Appendix E, this simple proof combines amplified privacy accountant by sub-sampling in Bun et al. (2018) with the tight composition theorem for Renyi DP (Mironov et al., 2019).

**Lemma 3.1** (Privacy Guarantee). *Provided that $B < 0.1N$, there exists absolute constants $c_1, c_2 > 0$ so that DP-SGD and DP-NSGD are $(\epsilon, \delta)$-differentially private for any $\epsilon < c_1 B^2 T/N^2$ and $\delta > 0$ if we choose $\sigma \geq c_2 \frac{\sqrt{T \log(1/\delta)}}{N\epsilon}$.*

### 3.2 Convergence Guarantee of DP-NSGD

For the DP-NSGD (Algorithm 1), we have the following convergence result.

**Theorem 3.2.** *Suppose that the objective $f(\boldsymbol{x})$ satisfies Assumption 2.1 and 2.2. Given any noise multiplier $\sigma$ and a regularizer $r > \tau_0$, we run DP-NSGD (Algorithm 1) using constant learning rate*

$$\eta = \sqrt{\frac{2}{(L_1(r + \tau_0) + L_0)Td\sigma^2}}, \tag{4}$$

*with sufficiently many iterations $T$ (larger than some constant determined by $(\sigma^2, d, L, \tau, r)$, as specified in Lemma C.2). We can obtain the following upper bound on gradient norm*

$$\mathbb{E}\left[\min_{0 \leq k < T} \|\nabla f(\boldsymbol{x}_k)\|\right] \leq C \left( \sqrt[4]{\frac{(D_f + 1)^2 r^3 d\sigma^2}{T}} + \sqrt[4]{\frac{1}{Tr^3 d\sigma^2}} \right) + \frac{8(r + 2\tau_0)\tau_0^2}{r(r + \tau_0)(1 - \tau_1)^3}, \tag{5}$$

*where $C$ is a constant depending on the gradient variance coefficients $(\tau_0, \tau_1)$ and the objective smoothness coefficients $(L_0, L_1)$.*

Theorem 3.2 is a general convergence for normalized SGD with perturbation. To achieve $(\epsilon, \delta)$-differential privacy, we can choose proper noise multiplier $\sigma$ and iterations $T$.

**Corollary 3.3.** *Under the same conditions of Theorem 3.2, we use $\sigma = c_2 \sqrt{T \log \frac{1}{\delta}}/(N\epsilon)$ with $c_2$ from Lemma 3.1 and set $T \geq \mathcal{O}(N^2\epsilon^2/(r^3 d \log \frac{1}{\delta}))$. If we have sufficiently many samples (larger than some constant determined by $(\epsilon, \delta, d, L, \tau, r, B)$, as specified in Lemma C.3), there holds the following privacy-utility trade-off*

$$\mathbb{E}\left[\min_{0 \leq k < T} \|\nabla f(\boldsymbol{x}_k)\|\right] \leq C' \sqrt[4]{\frac{dr^3 \log(1/\delta)}{N^2 \epsilon^2}} + \frac{8(r + 2\tau_0)\tau_0^2}{r(r + \tau_0)(1 - \tau_1)^3}, \tag{6}$$

*where $C'$ is a constant depending on the gradient variance coefficients $(\tau_0, \tau_1)$, the objective smoothness coefficients $(L_0, L_1)$, the function value gap $D_f$ and the batch size $B$.*

There are two major obstacles to prove this theorem. One is to handle the normalized gradients, which is solved by carefully using $r$ and dividing the range of $\|\nabla f(\boldsymbol{x}_k)\|$ into two cases. The other is to handle the Gaussian perturbation $\boldsymbol{z}$, whose variance $\sigma^2$ could even grow linearly with $T$. This is solved by setting the learning rate $\eta$ proportionally to $1/\sigma$ in (4). Combining two steps together, we reach the privacy-utility trade-off in Corollary 3.3.

*Proof Sketch of Theorem 3.2.* We firstly establish a descent inequality as in Lemma B.2 via exploiting the $(L_0, L_1)$-generalized smooth condition in Assumption 2.1,

$$\mathbb{E}_k [f(\boldsymbol{x}_{k+1})] - f(\boldsymbol{x}_k) \leq \underbrace{-\eta \mathbb{E}_k [\langle h_k \nabla f(\boldsymbol{x}_k), \boldsymbol{g}_k \rangle]}_{\mathfrak{A}} + \underbrace{\frac{L_0 + L_1 \|\nabla f(\boldsymbol{x}_k)\|}{2} \eta^2 \left(d\sigma^2 + \mathbb{E}_k \|h_k \boldsymbol{g}_k\|^2\right)}_{\mathfrak{B}}.$$

In the above expression, we use $\mathbb{E}_k$ to denote taking expectation of $\{\boldsymbol{g}_k^{(i)}, i \in \mathcal{S}_k\}$ and $\boldsymbol{z}_k$ conditioned on the past, especially $\boldsymbol{x}_k$. Next, in Lemma C.1, we upper bound the second order term $\mathfrak{B}$ by a constant $\mathcal{O}(\eta^2)$ plus a term like $\eta \mathbb{E}_k [h_k \|\nabla f(\boldsymbol{x}_k)\|^2]$, which is compatible to $\mathfrak{A}$. In order to find simplified lower bound for $\mathfrak{A}$, we separate the time index $\{0, 1, \cdots, T-1\}$ into two cases $\mathcal{U} := \left\{0 \leq k < T : \|\nabla f(\boldsymbol{x}_k)\| \geq \frac{\tau_0}{1 - \tau_1}\right\}$ and $\mathcal{U}^c$. Specifically, in Lemma B.3, we find that for $k \in \mathcal{U}$, the first order term $\mathfrak{A}$ is $\Omega(\eta\|\nabla f(\boldsymbol{x}_k)\|)$ (see (17) in Appendix B); for $k \notin \mathcal{U}$, the first order term $\mathfrak{A}$ is $\Omega(\eta(\|\nabla f(\boldsymbol{x}_k)\|^2/r - \tau_0^3/r^2))$ (see (18) in Appendix B). Then our result follows from summing up descent inequalities and scaling $\eta$ deliberately. □

There is rich literature investigating the convergence properties of *normalized gradient* methods in the non-private non-convex optimization setting. These results heavily rely on the following inequality to control the amount of descent

$$-\left\langle \nabla f(\boldsymbol{x}_k), \frac{\boldsymbol{g}_k}{\|\boldsymbol{g}_k\|} \right\rangle \leq -\|\nabla f(\boldsymbol{x}_k)\| + 2\|\boldsymbol{g}_k - \nabla f(\boldsymbol{x}_k)\|. \tag{7}$$

Based on this inequality, one unavoidably needs to control the error term $\|\boldsymbol{g}_k - \nabla f(\boldsymbol{x}_k)\|$ well to have the overall convergence. In practice, You et al. (2019) used large batch size $B \sim \mathcal{O}(T)$ to reduce the variance. However, this trick cannot be applied in the private setting due to *per-sample gradient processing*, e.g., clipping or normalization. Cutkosky & Mehta (2020) and Jin et al. (2021) use momentum techniques with a properly scaled weight decay and obtain a convergence rate $\mathbb{E}\|\nabla f(\boldsymbol{x}_T)\| = \mathcal{O}(1/\sqrt[4]{T})$, which is comparable with the usual SGD in the non-convex setup (Ghadimi & Lan, 2013). However, momentum techniques do not apply well in the private setting either, because we only have access to previous descent directions with noise due to the composite differential privacy requirement. As far as we know, there is no successful application of momentum in the private community, either practically or theoretically. Our analysis is able to deal with this issue, mainly because clipping / normalization is applied in a per-sample manner.

In this paper, we view the regularizer $r$ as a tunable hyperparameter, and make our upper bound decay as fast as possible $\mathcal{O}(1/\sqrt[4]{T})$ by tuning $r$. However, due to the restrictions imposed by privacy protection, we are unable to properly exploit large batch size or momentum to reduce variance as in previous works, thus leaving a strictly positive term $\mathcal{O}(\tau_0^2/r)$ in the right hand side of Theorem 3.2 and Corollary 3.3.

Another observation is that $r$ trades off between the non-vanishing bound $\mathcal{O}(\tau_0^2/r)$ and the decaying term $\mathcal{O}\left(\sqrt[4]{dr^3 \log \frac{1}{\delta}/(N^2\epsilon^2)}\right)$.

### 3.3 Convergence Guarantee of DP-SGD

We now turn our attention to DP-SGD with per-sample gradient clipping, whose convergence is given by the following theorem.

**Theorem 3.4.** *Suppose that the objective $f(\boldsymbol{x})$ satisfies Assumption 2.1 and 2.2. Given any noise multiplier $\sigma > 0$ and any clipping threshold $c > 2\tau_0/(1 - \tau_1)$, we run DP-SGD using constant learning rate*

$$\eta = \sqrt{\frac{2}{(L_1(c + \tau_0) + L_0)Tdc^2\sigma^2}}, \tag{8}$$

*with sufficiently many iterations $T$ (larger than some constant determined by $(\sigma^2, d, L, \tau, c)$ respectively, specified in Lemma D.2). We can obtain the following upper bound on gradient norms*

$$\mathbb{E}\left[\min_{0 \le k < T} \|\nabla f(\boldsymbol{x}_k)\|\right] \le C \left(\sqrt[4]{\frac{(D_f + 1)^2 c^3 d\sigma^2}{T}} + \sqrt[4]{\frac{1}{Tc^2(c + \tau_0)d\sigma^2}}\right), \tag{9}$$

*where we employ a constant $C$ only depending on the gradient variance coefficients $(\tau_0, \tau_1)$ and the objective smoothness coefficients $(L_0, L_1)$.*

Combining Theorem 3.4 and Lemma 3.1, we have a characterization for $(\epsilon, \delta)$-DP.

**Corollary 3.5.** *Under the same conditions of Theorem 3.4, we use $\sigma = c_2\sqrt{T\log(1/\delta)}/(N\epsilon)$ with $c_2$ from Lemma 3.1 and set $T \ge \mathcal{O}(N^2\epsilon^2/(c^3 d \log \frac{1}{\delta}))$. If we have sufficiently many samples (larger than some constant determined by $(\epsilon, \delta, d, L, \tau, c, B)$, as specified in Lemma D.3), there holds the following privacy-utility trade-off*

$$\mathbb{E}\left[\min_{0 \le k < T} \|\nabla f(\boldsymbol{x}_k)\|\right] \le C' \sqrt[4]{\frac{dc^3 \log(1/\delta)}{N^2\epsilon^2}}, \tag{10}$$

*where $C'$ is a constant depending on the gradient variance coefficients $(\tau_0, \tau_1)$, the objective smoothness coefficients $(L_0, L_1)$, the function value gap $D_f$ and the batch size $B$.*

By comparing Corollary 3.5 and Corollary 3.3, the most significant distinction of DP-SGD from DP-NSGD is that clipping does not induce a non-vanishing term $\mathcal{O}(\tau_0^2/r)$ as what we obtained in Corollary 3.3. This distinction is because $\mathfrak{A} := \eta\mathbb{E}\left[\langle\nabla f, h\boldsymbol{g}\rangle\right]$ and $\bar{\mathfrak{A}} := \eta\mathbb{E}\left[\langle\nabla f, \bar{h}\boldsymbol{g}\rangle\right]$ behave quite differently in some cases (see details in Lemmas B.3 & B.5 of Appendix B).

Specifically, when $\|\nabla f\|$ is larger than $\tau_0/(1 - \tau_1)$, we know $\langle\nabla f, \boldsymbol{g}\rangle \ge 0$. Therefore, the following ordering

$$\frac{c}{c + \|\boldsymbol{g}\|} \le \min\left\{1, \frac{c}{\|\boldsymbol{g}\|}\right\} \le \frac{2c}{c + \|\boldsymbol{g}\|} \tag{11}$$

guarantees $\mathfrak{A}$ and $\bar{\mathfrak{A}}$ to be equivalent to $\Omega(\eta\|\nabla f\|)$, where (11) can be argued by considering two cases $c > \|\boldsymbol{g}\|$ and $c \le \|\boldsymbol{g}\|$ separately. When the gradient norm $\|\nabla f\|$ is small, the inner-product $\langle\nabla f, \boldsymbol{g}\rangle$ could be of any sign, and we can only have $\mathfrak{A} = \Omega\left(\eta\left(\|\nabla f\|^2/r - \tau_0^3/r^2\right)\right)$ and $\bar{\mathfrak{A}} = \Omega\left(\eta\left(\|\nabla f\|^2\right)\right)$ instead. As $\mathfrak{A}$ controls the amount of descent within one iteration for DP-NSGD, the non-vanishing term appears. We here provide a toy example of the distribution of $\boldsymbol{g}$ to further illustrate the different behaviors of $\mathfrak{A}$ and $\bar{\mathfrak{A}}$.

**Example 3.1.** Consider a simple distribution on $\boldsymbol{e} \triangleq \boldsymbol{g} - \nabla f$:

$$\mathbb{P}\left(\boldsymbol{e} = \frac{\tau_0\nabla f}{\|\nabla f\|}\right) = \frac{1}{3}, \quad \mathbb{P}\left(\boldsymbol{e} = -\frac{\tau_0\nabla f}{2\|\nabla f\|}\right) = \frac{2}{3}.$$

This distribution certainly satisfies Assumption 2.2 with $\tau_1 = 0$. We calculate the explicit formula of $\mathfrak{A}$ for this case,

$$\mathfrak{A} = \frac{\eta(\|\nabla f\|^3 + (3r + \tau_0/2)\|\nabla f\|^2 - \tau_0^2\|\nabla f\|/2)}{3(r + \tau_0 + \|\nabla f\|)(r + \tau_0/2 - \|\nabla f\|)}.$$

For $\|\nabla f(\boldsymbol{x}_k)\| \leq \tau_0^2/(10r)$, we have $\mathfrak{A} < 0$. The function value may not decrease along $\mathbb{E}[h\boldsymbol{g}]$ in this case and the learning curves are expected to fluctuate adversely. This example also supports that the lower bound $\Omega\left(\eta\left(\|\nabla f\|^2/r - \tau_0^3/r^2\right)\right)$ on $\mathfrak{A}$ is optimal. In contrast, for the clipping operation, as long as $\|\nabla f\| \leq c - \tau_0$, we have $\bar{h} \equiv 1$ and therefore $\bar{\mathfrak{A}} = \eta\|\nabla f\|^2$.

As pointed out in the Example 3.1, the training trajectories of DP-NSGD fluctuate more adversely than DP-SGD, since $\mathfrak{A}$ can be of any sign while $\bar{\mathfrak{A}}$ stays positive. This difference is also observed empirically (see Figure 1) that the training loss of normalized SGD with $r = 0.01$ (close to normalization) fluctuates more than that of the clipped SGD with $c = 1$[3].

**Comparison to a concurrent work.** In the meantime of submitting our research to ArXiv, another group of researchers Bu et al. (2023) also achieved the same order of convergence rates but under markedly different assumptions. As displayed in Table 1, they use a stronger and more conventional notion of $L$-smoothness for the objective function. On the other hand, the assumptions on the gradient noise cannot be compared directly between theirs and ours. Our Assumption 2.2 is "stricter" than theirs in the sense that $\boldsymbol{g} - \nabla f$ is bounded almost surely, whereas Bu et al. (2023) only require a bounded second moment. However, they make an additional distributional assumption that $\boldsymbol{g}$ is centrally symmetric around its mean, which firstly appears in Chen et al. (2020) with supportive empirical evidence.

### 3.4 On the Biases from Normalization and Clipping

In this section, we further discuss how gradient *normalization* and *clipping* affect the overall convergences of the private algorithms. The influence is two-folded: one is that clipping/normalization induces *bias*, i.e., the gap between true gradient $\nabla f$ and clipped/normalized gradient; the other is that the added Gaussian noise for privacy may scale with the regularizer $r$ and the clipping threshold $c$.

**The Induced Bias.** When writing the objective as an empirical average $f(\boldsymbol{x}) = \sum_i \ell(\boldsymbol{x}, \xi_i)/N$, the true gradient is $\nabla f(\boldsymbol{x}) = \frac{1}{N}\sum_{i=1}^N \boldsymbol{g}^{(i)}$. Then both expected descent directions of Normalized SGD and Clipped SGD

$$\mathbb{E}[h\boldsymbol{g}] = \frac{1}{N}\sum_{i=1}^N \frac{\boldsymbol{g}^{(i)}}{r + \|\boldsymbol{g}^{(i)}\|}, \quad \mathbb{E}[\bar{h}\boldsymbol{g}] = \frac{1}{N}\sum_{i=1}^N \boldsymbol{g}^{(i)}\min\left\{1, \frac{c}{\|\boldsymbol{g}^{(i)}\|}\right\},$$

deviate from the true gradient $\nabla f(\boldsymbol{x})$. This means that the normalization or the clipping induces biases compared with the true gradient. A small regularizer $r$ or a small clipping threshold $c$ induces large *biases*, while a large $r$ or $c$ could reduce such *biases*. This can be seen from the training loss curves of different values of $c$ and $r$ in Figure 1, whose implementation details are in Section 4. Although theoretically the *bias* itself hinders convergence, the *biases* affect the accuracy curves differently for clipped SGD and normalized SGD. The accuracy curves of clipped SGD vary with the value of $c$ while the value of $r$ makes almost no impact on the accuracy curves of normalized SGD. This phenomenon extends to the private setting (look at the accuracy curves in Figure 2).

A qualitative explanation would be as follows. After several epochs of training, *well-fitted* samples $\xi$ (those already been classified correctly) yield small gradients $\boldsymbol{g}$ and *not-yet-fitted* samples $\xi'$ (those not been correctly classified) yield large gradients $\boldsymbol{g}'$. Typically, $c$ is set on the level of gradient norms, while $r$ is for regularizing the division. As the training goes, more and more samples are fitted well enough and their gradients would become small (He et al., 2023). While both normalized SGD and clipped SGD tend to amplify the significance of instances with small gradients (*well-fitted samples*), the amplification effect of normalized SGD is generally more pronounced than that of clipped SGD unless the clipping threshold is set to be extremely small. Moreover, normalized SGD achieves comparable accuracy but incurs higher loss than clipped SGD. We call for a future investigation towards understanding this phenomenon thoroughly. Specific to the setting $r \approx 0, c = 1$ in Figure 1, normalized SGD normalizes all $\boldsymbol{g}^{(i)}$ to be with a unit norm, while clipped SGD would not change gradients with small norms.

From a theoretical perspective, to give a finer-grained analysis of the *bias*, imposing further assumptions to control $\gamma$ may be a promising future direction. For example, Chen et al. (2020) made an attempt towards

---

[3] $c = 1$ makes the magnitude of clipped SGD similar as normalized SGD and hence the comparison is meaningful.

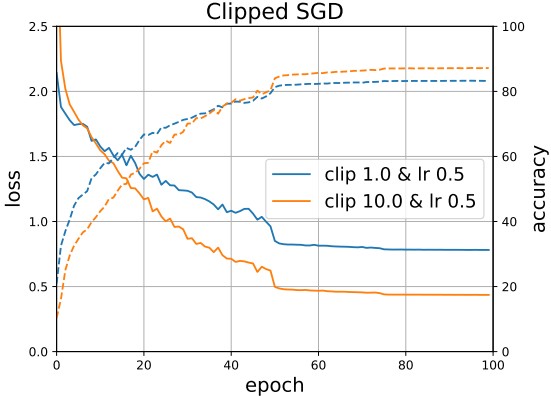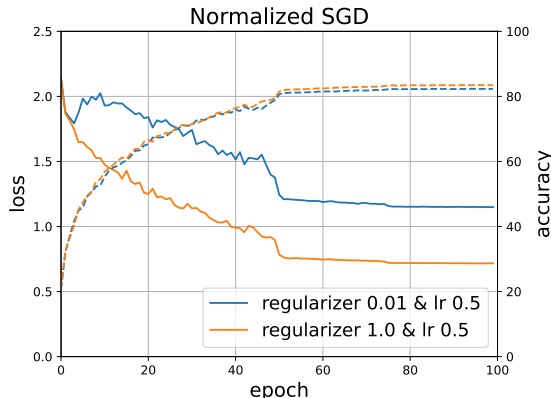

Figure 1: Left: Training loss and training accuracy curves of Clipped SGD. Right: Training loss and training accuracy curves of Normalized SGD. Both are trained with ResNet20 on CIFAR10 task.

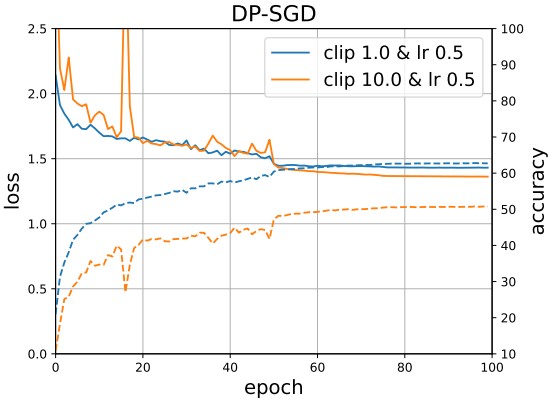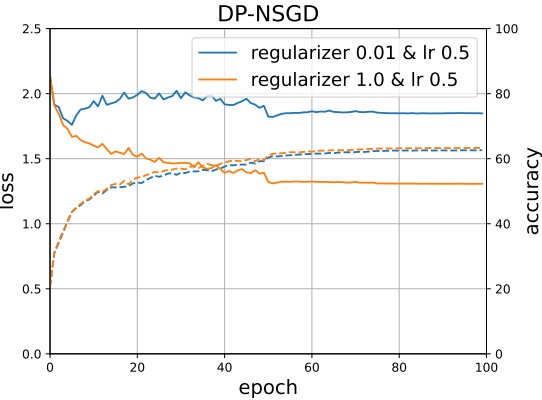

Figure 2: Left: Training loss and training accuracy curves of DP-SGD. Right: Training loss and training accuracy curves of DP-NSGD. Task: ResNet20 on CIFAR10 with $\epsilon = 8, \delta = 1\text{e-}5$.

this aspect, but their assumption is a bit artificial and not intuitive. Sankararaman et al. (2020) proposed a concept *gradient confusion*, defined as $\gamma = -\min\{\langle \boldsymbol{g}^{(i)}, \boldsymbol{g}^{(j)} \rangle : i \neq j\}$ to approximately quantify how per-sample gradients *align* to each other.

**The Added Noises for Privacy Guarantee**. For gradient *clipping*, the added noise (Gaussian perturbation) $\bar{z} \sim \mathcal{N}(0, c^2\sigma^2 \boldsymbol{I}_d)$ is proportional to $c$, while for gradient *normalizing* $z \sim \mathcal{N}(0, \sigma^2 \boldsymbol{I}_d)$ keeps invariant with $r$. This suggests that when tuning DP-SGD, $\eta$ needs to vary with $c$, in order to control the noise component $\eta\bar{z}$ in each update. In contrast, DP-NSGD is robust under different scales of $r$, and thus is easier to tune intuitively. Extensive experiments in Section 4 support this intuition empirically.

## 4    Experiments

This section conducts experiments to demonstrate the efficacy of Algorithm 1 and compare the behavior of DP-SGD and DP-NSGD empirically.

### 4.1    Tuning Vision Models

One example for the proof of concept is training a ResNet20 (He et al., 2016) with CIFAR-10 dataset. As in literature (Yousefpour et al., 2021; Davody et al., 2020; Yu et al., 2020), we replace all batch normalization layers with group normalization (Wu & He, 2018) layers for easily computing the per-sample gradients. The non-private accuracy for CIFAR-10 is 90.4%. We compare the performances of DP-NSGD and DP-SGD with a wide range of hyper-parameters and different learning rate scheduling rules. All experiments can be run on a single Tesla V100 with 16GB memory. The ResNet20 has 270K trainable parameters.

**Hyperparameter choices.** We first fix the privacy budget $\epsilon = \{2.0, 4.0, 8.0\}, \delta = 10^{-5}$, which corresponds to setting the noise multiplier $\sigma = \{3.6, 2.0, 1.2\}$ for the case of batch size 1000 and number of epochs 100 with Rényi differential privacy accountant (Abadi et al., 2016; Mironov et al., 2019). There are tighter privacy accountants (Gopi et al., 2021) that can save $\epsilon$ for the same noise multiplier. We then fix the weight decay to be 0 and use the classical learning rate scheduling strategy that multiplies the initial $lr$ with 0.1 at epoch 50 and 0.01 at epoch 75 respectively. The hyperparameters to tune are the initial learning rate $lr$ and the clip threshold $c$ for DP-SGD, where $lr$ takes values $\{0.05, 0.1, 0.2, 0.4, 0.8, 1.6, 3.2\}$ and $c$ takes values $\{0.1, 0.4, 1.6, 6.4, 12.8\}$. At the same time, the hyperparameters to tune for DP-NSGD are the initial learning rate $lr$ and the regularizer $r$, where $lr$ takes values $\{0.05, 0.1, 0.2, 0.4, 0.8, 1.6, 3.2\}$ and $r$ takes values $\{0.0001, 0.001, 0.01, 0.1, 1.0\}$. We compare the validation accuracy of DP-SGD and DP-NSGD via heatmaps of the above hyperparameter choices in Figure 3. We can see that the performance of DP-NSGD is rather stable for the regularizer taking values from $10^{-4}$ to 1.0 and it is mostly affected by the learning rate. This is in sharp contrast with the case of DP-SGD where the performance depends on both the learning rate and the clip threshold in a complicated way. This suggests that it may be easier to tune the hyperparameters for DP-NSGD than that for DP-SGD, which may help save the privacy budget for tuning hyperparameters (Papernot & Steinke, 2021).

We also run the above setting with the cyclic learning rate scheduling with min-lr = 0.02 and max-lr = 1.0. The best accuracy number are of DP-NSGD and DP-SGD can be as good as 66, which is comparable with the number achieved with model architecture modification in Papernot et al. (2020b).

### 4.2    Fine-tuning Large Language Models

We use the pretrained RoBERTa model (Liu et al., 2019)[④], which has 125M parameters (RoBERTa-Base) and fine-tune them except the embedding layer for SST-2 classification task (Wang et al., 2018). We adopt the setting as in Li et al. (2021): full-precision training with the batch size 1000 and the number of epochs 10.

**Hyperparameter choice:** For privacy parameters, we use $\epsilon = 8, \delta =$ 1e-5. With Renyi differential privacy accountant, this corresponds to setting the noise multiplier 0.635. We compare the behavior of DP-SGD

---

[④]The model and checkpoints can be found at https://github.com/pytorch/fairseq/tree/master/examples/roberta.

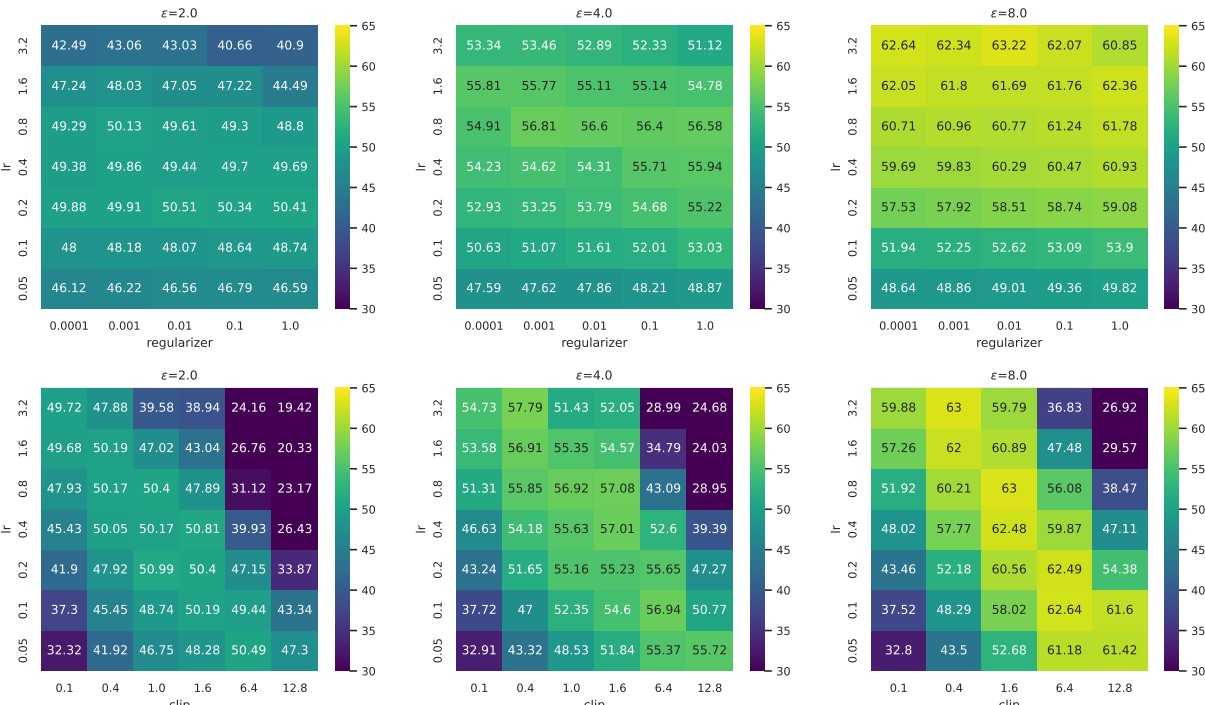

Figure 3: Experiments for ResNet20 on CIFAR10 task. Upper: Accuracy heatmap of DP-NSGD with varying lrs and regularizers. Lower: Accuracy heatmap of DP-SGD with varying lrs and clipping thresholds. The DP parameters are $\delta = 1e^{-5}$ and $\epsilon = 2.0, 4.0, 8.0$ from left to right.

and that of DP-NSGD. For DP-SGD, we search the clipping threshold $c$ from $\{0.1, 0.5, 2.5, 12.5, 50.0\}$ and the $lr$ from $\{0.05, 0.1, 0.2, 0.4, 0.8, 1.6\}$. For the DP-NSGD, we search the learning rate $lr$ over the same set of DP-SGD and the regularizer $r$ from $\{$1e-3, 1e-2, 1e-1, 1., 10.0$\}$.

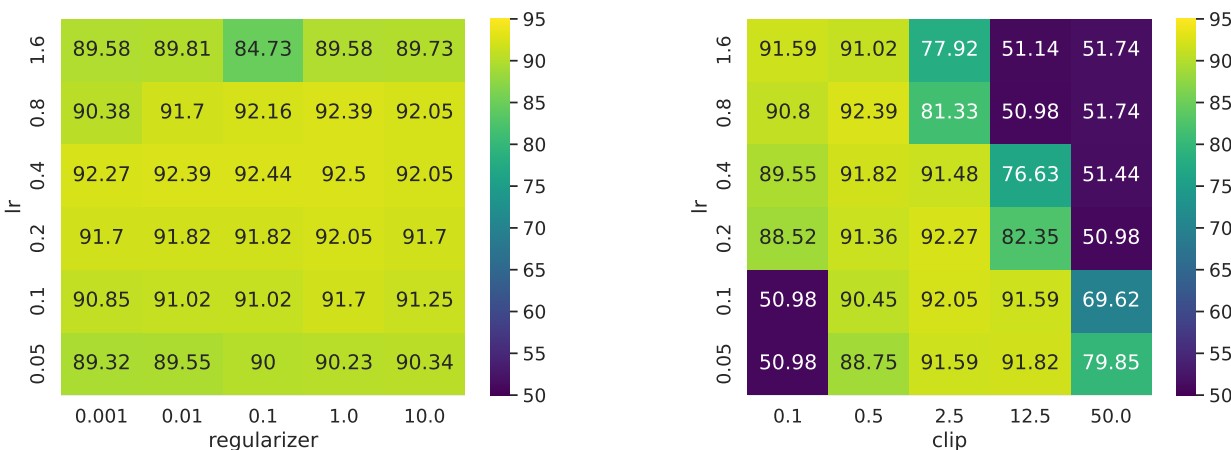

Figure 4: Experiments of fine-tuning RoBERTa on SST-2 task. Left: Accuracy heatmap of DP-NSGD with varying learning rates and regularizers. Right: Accuracy heatmap of DP-SGD with varying learning rates and clipping thresholds.

We have similar observation in Figure 4 that the performance of DP-NSGD is rather stable for the regularizer and the learning rate, which indicates that it could be easier to tune than DP-SGD.

Notably, the concurrent study by Bu et al. (2023) has conducted extensive experiments, supporting observations that both DP-SGD and DP-NSGD achieve comparable performance. Furthermore, it was also found that DP-NSGD is comparatively easier to tune.

## 5 Concluding Remarks

In this paper, we have studied the convergence of two algorithms, i.e., DP-SGD and DP-NSGD, for differentially private non-convex empirical risk minimization. We have achieved a rate that significantly improves over previous literature under similar setup and have analyzed the bias induced by the clipping or normalizing operation. As for future directions, it is very interesting to consider the convergence theorems under stronger assumptions on the gradient distribution.

### Acknowledgments

All acknowledgements go at the end of the paper before appendices and references. Moreover, you are required to declare funding (financial activities supporting the submitted work) and competing interests (related financial activities outside the submitted work). More information about this disclosure can be found on the TMLR website.

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

# Appendices

## A   More Literature on Private Optimization

**Private Non-convex Empirical Risk Minimization:** This line of works (Wang et al., 2019a; 2023; Zhou et al., 2020) study GD, RMSprop and Adam for centralized differentially private non-convex empirical risk minimization $\min_{\boldsymbol{x}} f(\boldsymbol{x}) = \sum_{i=1}^{N} \ell(\boldsymbol{x}, \xi_i)/N$. As our Table 1 suggests, all these algorithms achieve the state-of-the-art on the utility upper bound $\mathcal{O}\left(\sqrt[4]{d/(N^2\epsilon^2)}\right)$, under the following assumption.

**Assumption A.1** (Previous assumption for non-convex DP ERM)**.** *There exists $L, G > 0$ such that for any $\xi$, the loss function $\boldsymbol{x} \mapsto \ell(\boldsymbol{x}, \xi)$ is $L$-smooth ($\|\nabla_{\boldsymbol{x}}^2 \ell(\boldsymbol{x}, \xi)\| \leq L$) and $G$-Lipschitz ($\|\nabla_{\boldsymbol{x}} \ell(\boldsymbol{x}, \xi)\| \leq G$).*

In direct comparison, our assumptions are much weaker:

- Assumption 2.1 only requires the expected loss function $f(\boldsymbol{x})$ to satisfy certain smoothness condition;

- Assumption 2.2 with $\tau_0 = 2G, \tau_1 = 0$ covers the cases of Assumption A.1.

In these works, the Lipschitz condition of the loss gradient $\nabla_{\boldsymbol{x}} \ell(\boldsymbol{x}, \xi)$ is vital since it allows to ignore the effect of gradient clipping when analyzing convergence, thus unable to provide theoretical understanding towards this tunable hyper parameter.

We manage to obtain best available utility bounds for the problem of non-convex DP ERM, even under weakened assumptions. More importantly, our weakened assumptions are not only closer to real-life neural network training, but also more suitable to show the distinctions between DP-SGD and DP-NSGD, as shown in Corollaries 3.5 & 3.3.

**Private Convex Optimization with Heavy-tailed Data:** In comparison with DP-ERM, *private convex optimization* (DP-SCO) (Bassily et al., 2014) privately minimizes the *population* risk

$$\min_{\boldsymbol{x}} \tilde{f}(\boldsymbol{x}) := \mathbb{E}_{\xi \sim \mathcal{P}}[\ell(\boldsymbol{x}, \xi)],$$

given i.i.d. samples $\xi_1, \ldots, \xi_N \sim \mathcal{P}$, where $\mathcal{P}$ is the underlying true distribution not solely the empirical one. Typically, the loss $\boldsymbol{x} \mapsto \ell(\boldsymbol{x}, \xi)$ is assumed to be convex for any $\xi$. For this problem, the utility is thus measured by $\mathbb{E}_{\xi_1^N, \mathcal{M}}[\tilde{f}(\boldsymbol{x}_{\text{output}})]$, where expectation is taken over both the dataset and the algorithm itself.

Many papers have investigated the problem of DP-SCO (Chaudhuri & Monteleoni, 2009; Wang et al., 2017; Kuru et al., 2020; Yu et al., 2020; Asi et al., 2021), but all of them adopt the Lipschitz condition on the loss gradient $\nabla_{\boldsymbol{x}} \ell(\boldsymbol{x}, \xi)$ to avoid considering gradient clipping. In contrast to our Assumption 2.2, recent progresses (Wang et al., 2020; Kamath et al., 2022; Hu et al., 2022) in DP SCO, weakened the Lipschitz condition to a heavy-tailed assumption, namely the $k$-th bounded moment condition on each coordinate $\nabla_{\boldsymbol{x}} \ell(\boldsymbol{x}, \xi)_j, j \in [d]$ of the gradient estimation (for example, (Kamath et al., 2022, Definition 2.11)). These works still have to assume the expected gradients to be uniformly bounded $\mathbb{E}_{\xi \sim \mathcal{P}}[\nabla_{\boldsymbol{x}} \ell(\boldsymbol{x}, \xi)]$ for every $\boldsymbol{x}$, e.g. the 6th assertion in (Kamath et al., 2022, Assumption 2.11). In contrast, our setting is able to illustrate the differences of clipping and normalizing based methods.

From an algorithmic perspective, Wang et al. (2020); Kamath et al. (2022); Hu et al. (2022) proposed methods involving a sophisticated mean oracle in Kamath et al. (2020) to estimate the expected gradients, while this mean oracle still employs clipping with respect to a preset threshold. Specifically, to handle the ill-behaved gradients, methods from Kamath et al. (2022) process each coordinate separately, by firstly partitioning the selected batch and then taking median over the clipped means of each disjoint parts. These techniques might be helpful in addressing non-convex DP ERM with heavy-tailed conditions.

**Private Federated Learning** In recent years, private federated learning also draws lots of attention. Since our work focuses on *centralized* DP, we only mention a few related works in this direction. Das et al.

(2021) studied DP-NormFedAvg, a client-level DP optimizer in a federated setting. Their optimizer, DP-NormFedAvg, uses vanilla GD for each client and normalizes the contribution of every client to unit-norm. Sharing similar motivation with our centralized DP-NSGD, their contributions are roughly credited to DP-NSGD. Their convergence analysis is based on one-point/quasar convexity and $L$-smoothness. Specifically for the bound shown in the 4th line of Table 1, $\|\boldsymbol{x}^{(i)} - \boldsymbol{x}^*\|$ measures heterogeneity via the distance between the $i$-th client's local minimizer $\boldsymbol{x}^{(i)}$ to the global minimizer $\boldsymbol{x}^*$, and $D_X \triangleq \|\boldsymbol{x}_0 - \boldsymbol{x}^*\|$.

Following Chen et al. (2020), DP-FedAvg with clipping is analyzed in Zhang et al. (2021), with symmetric gradient distribution assumption. Additionally, a recent preprint (Lowy et al., 2022) also investigates private non-convex federated learning, but based on local/shuffle differential privacy.

## B   Prerequisite Lemmas

The following is a standard lemma for $(L_0, L_1)$-generalized smooth functions, and it can be obtained via Taylor's expansion. Throughout this appendix, we use $\mathbb{E}_k$ to denote taking expectation of $\{\boldsymbol{g}_k^{(i)}, i \in \mathcal{S}_k\}$ and $\boldsymbol{z}_k$ conditioned on the past, especially $\boldsymbol{x}_k$.

**Lemma B.1** (Lemma C.4, Jin et al. (2021)). *A function $f : \mathbb{R}^d \to \mathbb{R}^d$ is $(L_0, L_1)$-generalized smooth, then for any $\boldsymbol{x}, \boldsymbol{x}^+ \in \mathbb{R}^d$,*

$$f(\boldsymbol{x}^+) \le f(\boldsymbol{x}) + \langle \nabla f(\boldsymbol{x}), \boldsymbol{x}^+ - \boldsymbol{x} \rangle + \frac{L_0 + L_1 \|\nabla f(\boldsymbol{x})\|}{2} \|\boldsymbol{x}^+ - \boldsymbol{x}\|^2.$$

**Lemma B.2.** *For any $k \ge 0$, we use $\boldsymbol{g}_k$ to denote another realization of the underlying distribution behind the set of i.i.d. unbiased estimates $\{\boldsymbol{g}_k^{(i)} : i \in \mathcal{S}_k\}$. If we run DP-NSGD iteratively, the trajectory would satisfy the following bound:*

$$\mathbb{E}_k \left[ f(\boldsymbol{x}_{k+1}) \right] - f(\boldsymbol{x}_k) \le -\eta \mathbb{E}_k \left[ \langle h_k \nabla f(\boldsymbol{x}_k), \boldsymbol{g}_k \rangle \right] + \frac{L_0 + L_1 \|\nabla f(\boldsymbol{x}_k)\|}{2} \eta^2 \left( d\sigma^2 + \mathbb{E}_k \|h_k \boldsymbol{g}_k\|^2 \right). \tag{12}$$

*Proof.* The updating rule of our iterative algorithm could be summarized as

$$\boldsymbol{x}_{k+1} = \boldsymbol{x}_k - \eta \left( \frac{1}{B} \sum_{i \in \mathcal{S}_k} h_k^{(i)} g_k^{(i)} + \boldsymbol{z}_k \right), \quad \boldsymbol{z}_k \sim \mathcal{N}(0, \sigma^2 \boldsymbol{I}_d).$$

By taking expectation $\mathbb{E}_k$ conditioned on the past, we rewrite the first-order term in Lemma B.1 into

$$\mathbb{E}_k \left[ \langle \nabla f(\boldsymbol{x}), \boldsymbol{x}_k - \boldsymbol{x}_{k+1} \rangle \right] = \eta \mathbb{E}_k \left[ \langle h_k \nabla f(\boldsymbol{x}_k), \boldsymbol{g}_k \rangle \right]. \tag{13}$$

In the same manner, we bound the second-order term by

$$\mathbb{E}_k \|\boldsymbol{x}_{k+1} - \boldsymbol{x}_k\|^2 = \eta^2 d\sigma^2 + \frac{\eta^2}{B^2} \mathbb{E}_k \left\| \sum_{i \in \mathcal{S}_k} h_k^{(i)} \boldsymbol{g}_k^{(i)} \right\|^2 \le \eta^2 \left( d\sigma^2 + \mathbb{E}_k \|h_k \boldsymbol{g}_k\|^2 \right), \tag{14}$$

where the last inequality follows from an elementary Cauchy-Schwarz inequality,

$$\left\| \sum_{i \in \mathcal{S}_k} h_k^{(i)} \boldsymbol{g}_k^{(i)} \right\|^2 \le B \sum_{i \in \mathcal{S}_k} \left\| h_k^{(i)} \boldsymbol{g}_k^{(i)} \right\|^2.$$

Plug (13) and (14) into Lemma B.1 to obtain the desired result. $\qquad \square$

*Remark* B.1. This lemma implies that mini batch size $B$ does not affect expected upper bounds, due to *per-sample* gradient normalization. We need to point out that $B$ could still influence high-probability upper bounds, and call for future investigations.

In Section C.1, we will upper bound the second-order term $\frac{L_0 + L_1 \|\nabla f(\boldsymbol{x}_k)\|}{2} \eta^2 \left( d\sigma^2 + \mathbb{E}_k \|h_k \boldsymbol{g}_k\|^2 \right)$ by a sum of $\alpha \eta h_k \|\nabla f(\boldsymbol{x}_k)\|$ (for some $0 < \alpha < 1$) and another term of $\mathcal{O}(\eta^2)$ via a proper scaling of $\eta$. We firstly present the following lemma to provide a simplified lower bound for the first-order terms

$$\eta \mathbb{E}_k \left[ \langle h_k \nabla f(\boldsymbol{x}_k), \boldsymbol{g}_k \rangle \right] - \eta \alpha \mathbb{E}_k[h_k] \|\nabla f(\boldsymbol{x}_k)\|^2. \tag{15}$$

**Lemma B.3** (Lower bound first-order terms for normalizing). *Define a function* $A : \mathbb{R}_+ \to \mathbb{R}$ *as*

$$A(s) = \begin{cases} \left( \dfrac{\tau_0}{r(1 - \tau_1) + 2\tau_0} - \dfrac{\alpha}{1 - \tau_1} \right) s, & \text{if } s \geq \dfrac{\tau_0}{1 - \tau_1}; \\ \dfrac{(1 - \alpha)(1 - \tau_1)}{r(1 - \tau_1) + 2\tau_0} s^2 - \dfrac{4\tau_0^3}{r(r + \tau_0)(1 - \tau_1)^3}, & \text{otherwise.} \end{cases} \tag{16}$$

*Then we have*

$$\eta \mathbb{E}_k \left[ \langle h_k \nabla f(\boldsymbol{x}_k), \boldsymbol{g}_k \rangle \right] - \eta \alpha \mathbb{E}_k[h_k] \|\nabla f(\boldsymbol{x}_k)\|^2 \geq \eta A(\|\nabla f(\boldsymbol{x}_k)\|).$$

*Proof.* We prove this lemma via separating the range of $\|\nabla f(\boldsymbol{x}_k)\|$. When $\|\nabla f(\boldsymbol{x}_k)\| \geq \tau_0/(1 - \tau_1)$, then

$$\langle \nabla f(\boldsymbol{x}_k), \boldsymbol{g}_k \rangle = \|\nabla f(\boldsymbol{x}_k)\|^2 + \langle \nabla f(\boldsymbol{x}_k), \boldsymbol{g}_k - \nabla f(\boldsymbol{x}_k) \rangle$$
$$\geq (1 - \tau_1) \|\nabla f(\boldsymbol{x}_k)\|^2 - \tau_0 \|\nabla f(\boldsymbol{x}_k)\| \geq 0,$$

followed by

$$\eta \mathbb{E}_k \left[ \langle h_k \nabla f(\boldsymbol{x}_k), \boldsymbol{g}_k \rangle \right] - \eta \alpha \mathbb{E}_k[h_k] \|\nabla f(\boldsymbol{x}_k)\|^2$$
$$= \mathbb{E}_k \left[ \frac{\eta \langle \nabla f(\boldsymbol{x}_k), \boldsymbol{g}_k \rangle}{r + \|\boldsymbol{g}_k\|} \right] - \mathbb{E}_k \left[ \frac{\alpha \eta}{(r + \|\boldsymbol{g}_k\|)} \|\nabla f(\boldsymbol{x}_k)\|^2 \right]$$
$$\geq \mathbb{E}_k \left[ \frac{\eta \langle \nabla f(\boldsymbol{x}_k), \boldsymbol{g}_k \rangle}{r + \tau_0 + (1 + \tau_1) \|\nabla f(\boldsymbol{x}_k)\|} \right] - \frac{\alpha}{1 - \tau_1} \eta \|\nabla f(\boldsymbol{x}_k)\|$$
$$= \frac{\eta \|\nabla f(\boldsymbol{x}_k)\|^2}{r + \tau_0 + (1 + \tau_1) \|\nabla f(\boldsymbol{x}_k)\|} - \frac{\alpha}{1 - \tau_1} \eta \|\nabla f(\boldsymbol{x}_k)\|$$
$$\geq \left( \frac{\tau_0}{r(1 - \tau_1) + 2\tau_0} - \frac{\alpha}{1 - \tau_1} \right) \eta \|\nabla f(\boldsymbol{x}_k)\|. \tag{17}$$

When $\|\nabla f(\boldsymbol{x}_k)\| < \tau_0/(1 - \tau_1)$, we have $\|\boldsymbol{g}_k - \nabla f(\boldsymbol{x}_k)\| \leq \tau_0 + \tau_1 \|\nabla f(\boldsymbol{x}_k)\| \leq \tau_0/(1 - \tau_1)$ as well. Then we decompose the first-order terms by

$$\eta \mathbb{E}_k \left[ \langle h_k \nabla f(\boldsymbol{x}_k), \boldsymbol{g}_k \rangle \right] - \eta \alpha \mathbb{E}_k[h_k] \|\nabla f(\boldsymbol{x}_k)\|^2$$
$$= (1 - \alpha) \eta \mathbb{E}_k[h_k] \|\nabla f(\boldsymbol{x}_k)\|^2 + \mathbb{E}_k \left[ \frac{\eta \langle \nabla f(\boldsymbol{x}_k), \boldsymbol{g}_k - \nabla f(\boldsymbol{x}_k) \rangle}{r + \|\boldsymbol{g}_k\|} \right].$$

On one hand, we know

$$h_k = \frac{1}{r + \|\boldsymbol{g}_k\|} \geq \frac{1}{r + \tau_0 + (1 + \tau_1) \|\nabla f(\boldsymbol{x}_k)\|} \geq \frac{1 - \tau_1}{r(1 - \tau_1) + 2\tau_0}.$$

On the other hand, we also have

$$\mathbb{E}_k \left[ \frac{\eta \langle \nabla f(\boldsymbol{x}_k), \boldsymbol{g}_k - \nabla f(\boldsymbol{x}_k) \rangle}{r + \|\boldsymbol{g}_k\|} \right]$$
$$= \mathbb{E}_k \left[ \frac{\eta \langle \nabla f(\boldsymbol{x}_k), \boldsymbol{g}_k - \nabla f(\boldsymbol{x}_k) \rangle}{r + \tau_0 + (1 + \tau_1) \|\nabla f(\boldsymbol{x}_k)\|} \right] + \mathbb{E}_k \left[ \frac{\eta \left[ (1 + \tau_1) \|\nabla f(\boldsymbol{x}_k)\| + \tau_0 - \|\boldsymbol{g}_k\| \right] \langle \nabla f(\boldsymbol{x}_k), \boldsymbol{g}_k - \nabla f(\boldsymbol{x}_k) \rangle}{(r + \|\boldsymbol{g}_k\|)(r + \tau_0 + (1 + \tau_1) \|\nabla f(\boldsymbol{x}_k)\|)} \right]$$
$$\geq -\eta \frac{4\tau_0^3}{r(r + \tau_0)(1 - \tau_1)^3}.$$

Therefore, we have

$$\eta \mathbb{E}_k \left[ \langle h_k \nabla f(\boldsymbol{x}_k), \boldsymbol{g}_k \rangle \right] - \eta \alpha \mathbb{E}_k [h_k] \|\nabla f(\boldsymbol{x}_k)\|^2$$
$$\geq \eta \left[ \frac{(1-\alpha)(1-\tau_1)}{r(1-\tau_1) + 2\tau_0} \|\nabla f(\boldsymbol{x}_k)\|^2 - \frac{4\tau_0^3}{r(r+\tau_0)(1-\tau_1)^3} \right]. \tag{18}$$

Combine both cases to derive the required lemma. □

We then establish similar results for gradient clipping based algorithms, whose updating rule is described by (2). Firstly, we mimic the proof of Lemma B.2 and derive without proof the following lemma.

**Lemma B.4.** *For any $k \geq 0$, we use $\boldsymbol{g}_k$ to denote another realization of the underlying distribution behind the set of i.i.d. unbiased estimates $\{\boldsymbol{g}_k^{(i)} : i \in \mathcal{S}_k\}$. If we run DP-SGD iteratively, the trajectory would satisfy the following bound:*

$$\mathbb{E}_k \left[ f(\boldsymbol{x}_{k+1}) \right] - f(\boldsymbol{x}_k) \leq -\eta \mathbb{E}_k \left[ \langle \bar{h}_k \nabla f(\boldsymbol{x}_k), \boldsymbol{g}_k \rangle \right] + \frac{L_0 + L_1 \|\nabla f(\boldsymbol{x}_k)\|}{2} \eta^2 \left( dc^2 \sigma^2 + \mathbb{E}_k \left\| \bar{h}_k \boldsymbol{g}_k \right\|^2 \right). \tag{19}$$

Then, we provide another lemma for clipping, similar to Lemma B.3.

**Lemma B.5** (Lower bound first-order terms for clipping). *Define a function $B : \mathbb{R}_+ \to \mathbb{R}$ as*

$$B(s) = \begin{cases} \left( \dfrac{\tau_0 c}{c(1-\tau_1) + 2\tau_0} - \dfrac{\alpha}{1-\tau_1} \right) s, & \text{if } s \geq \dfrac{\tau_0}{1-\tau_1}; \\ (1-\alpha)s^2, & \text{otherwise.} \end{cases} \tag{20}$$

*If we take $c \geq 2\tau_0/(1-\tau_1)$, then*

$$\eta \mathbb{E}_k \left[ \langle \bar{h}_k \nabla f(\boldsymbol{x}_k), \boldsymbol{g}_k \rangle \right] - \eta \alpha \mathbb{E}_k [\bar{h}_k] \|\nabla f(\boldsymbol{x}_k)\|^2 \geq \eta B(\|\nabla f(\boldsymbol{x}_k)\|).$$

*Proof.* Recall that $\bar{h}_k$ is defined as

$$\bar{h}_k = \min \left\{ 1, \frac{c}{\|\boldsymbol{g}_k\|} \right\} \geq \frac{c}{c + \|\boldsymbol{g}_k\|}.$$

Again, we take the strategy of separating the range of $\|\nabla f(\boldsymbol{x}_k)\|$. When $\|\nabla f(\boldsymbol{x}_k)\| \geq \tau_0/(1-\tau_1)$, we know $\langle \nabla f(\boldsymbol{x}_k), \boldsymbol{g}_k \rangle \geq 0$, followed by

$$\eta \mathbb{E}_k \left[ \langle \bar{h}_k \nabla f(\boldsymbol{x}_k), \boldsymbol{g}_k \rangle \right] - \eta \alpha \mathbb{E}_k [\bar{h}_k] \|\nabla f(\boldsymbol{x}_k)\|^2$$
$$\geq \mathbb{E}_k \left[ \frac{\eta c \langle \nabla f(\boldsymbol{x}_k), \boldsymbol{g}_k \rangle}{c + \|\boldsymbol{g}_k\|} \right] - \mathbb{E}_k \left[ \frac{\alpha \eta}{\|\boldsymbol{g}_k\|} \|\nabla f(\boldsymbol{x}_k)\|^2 \right]$$
$$\geq \mathbb{E}_k \left[ \frac{\eta c \langle \nabla f(\boldsymbol{x}_k), \boldsymbol{g}_k \rangle}{c + \tau_0 + (1+\tau_1)\|\nabla f(\boldsymbol{x}_k)\|} \right] - \frac{\alpha}{1-\tau_1} \eta \|\nabla f(\boldsymbol{x}_k)\|$$
$$= \frac{\eta c \|\nabla f(\boldsymbol{x}_k)\|^2}{c + \tau_0 + (1+\tau_1)\|\nabla f(\boldsymbol{x}_k)\|} - \frac{\alpha}{1-\tau_1} \eta \|\nabla f(\boldsymbol{x}_k)\|$$
$$\geq \left( \frac{\tau_0 c}{c(1-\tau_1) + 2\tau_0} - \frac{\alpha}{1-\tau_1} \right) \eta \|\nabla f(\boldsymbol{x}_k)\|.$$

Otherwise, when $\|\nabla f(\boldsymbol{x}_k)\| < \tau_0/(1-\tau_1) \leq (c-\tau_0)/(1+\tau_1)$ where the second inequality follows from $c \geq 2\tau_0/(1-\tau_1)$, we know

$$\|\boldsymbol{g}_k\| \leq \tau_0 + (1+\tau_1)\|\nabla f(\boldsymbol{x}_k)\| \leq c.$$

Therefore in this case, $\bar{h}_k = 1$ and

$$\eta \mathbb{E}_k \left[ \langle \bar{h}_k \nabla f(\boldsymbol{x}_k), \boldsymbol{g}_k \rangle \right] - \eta \alpha \mathbb{E}_k [\bar{h}_k] \|\nabla f(\boldsymbol{x}_k)\|^2 = (1-\alpha)\eta \|\nabla f(\boldsymbol{x}_k)\|^2.$$

Combine both cases to conclude the desired lemma. □

## C  Proofs for DP-NSGD

In this section, we provide a rigorous convergence theory for *normalized stochastic gradient descent with perturbation*. An unavoidable error between $\boldsymbol{g}_k$ and $\nabla f(\boldsymbol{x}_k)$ is a central distinction between stochastic and deterministic optimization methods. We begin with an explicit decomposition for (12)

$$
\begin{aligned}
\mathbb{E}_k & \left[ f(\boldsymbol{x}_{k+1}) \right] - f(\boldsymbol{x}_k) \\
&\leq -\eta \mathbb{E}_k \left[ h_k \right] \| \nabla f(\boldsymbol{x}_k) \|^2 - \eta \mathbb{E}_k \left[ \langle h_k \nabla f(\boldsymbol{x}_k), \boldsymbol{g}_k - \nabla f(\boldsymbol{x}_k) \rangle \right] \\
&\quad + \frac{L_0 + L_1 \| \nabla f(\boldsymbol{x}_k) \|}{2} \eta^2 \left( \mathbb{E}_k \left[ h_k^2 \| \nabla f(\boldsymbol{x}_k) \|^2 \right] + 2 \mathbb{E}_k \left[ h_k^2 \langle \boldsymbol{g}_k - \nabla f(\boldsymbol{x}_k), \nabla f(\boldsymbol{x}_k) \rangle \right] \right) \\
&\quad + \frac{L_0 + L_1 \| \nabla f(\boldsymbol{x}_k) \|}{2} \eta^2 \left( d\sigma^2 + \mathbb{E}_k \left[ h_k^2 \| \boldsymbol{g}_k - \nabla f(\boldsymbol{x}_k) \|^2 \right] \right).
\end{aligned}
\tag{21}
$$

### C.1  Upper Bound Second-order Terms

In theory, we need to carefully distinguish these terms accourding to their orders of $\eta$, as the first order term $\mathbb{E}_k \left[ \langle h_k \nabla f(\boldsymbol{x}_k), \boldsymbol{g}_k \rangle \right]$ controls the amount of descent mainly. We show that the second order terms could be bounded by first order terms via a proper scaling of $\eta$, in the following technical lemma.

**Lemma C.1.** *For any $0 < \alpha < 1$ to be determined explicitly later, if*

$$
\eta \leq \min \left( \frac{(r - \tau_0)\alpha}{4 L_0}, \frac{(1 - \tau_1)\alpha}{4 L_1}, \frac{\alpha}{6 L_1 d\sigma^2} \right)
\tag{22}
$$

*then we have*

$$
\frac{L_0 + L_1 \| \nabla f(\boldsymbol{x}_k) \|}{2} \eta^2 d\sigma^2 \leq \frac{L_0 + L_1(r + \tau_0)}{2} \eta^2 d\sigma^2 + \frac{\alpha \eta h_k}{4} \| \nabla f(\boldsymbol{x}_k) \|^2,
\tag{23}
$$

$$
(L_0 + L_1 \| \nabla f(\boldsymbol{x}_k) \|) \eta^2 h_k^2 \langle \nabla f(\boldsymbol{x}_k), \boldsymbol{g}_k - \nabla f(\boldsymbol{x}_k) \rangle \leq \frac{(L_0(1 - \tau_1) + L_1 \tau_0)\tau_0^2}{r^2(1 - \tau_1)^3} \eta^2 + \frac{\alpha \eta h_k}{4} \| \nabla f(\boldsymbol{x}_k) \|^2,
\tag{24}
$$

$$
\frac{L_0 + L_1 \| \nabla f(\boldsymbol{x}_k) \|}{2} \eta^2 h_k^2 \| \boldsymbol{g}_k - \nabla f(\boldsymbol{x}_k) \|^2 \leq \frac{(L_0(1 - \tau_1) + L_1 \tau_0)\tau_0^2}{2 r^2(1 - \tau_1)^3} \eta^2 + \frac{\alpha \eta h_k}{4} \| \nabla f(\boldsymbol{x}_k) \|^2,
\tag{25}
$$

$$
\frac{L_0 + L_1 \| \nabla f(\boldsymbol{x}_k) \|}{2} \eta^2 h_k^2 \| \nabla f(\boldsymbol{x}_k) \|^2 \leq \frac{\alpha \eta h_k}{4} \| \nabla f(\boldsymbol{x}_k) \|^2.
\tag{26}
$$

*Remark* C.1. These bounds are proved by separating the range of $\| \nabla f(\boldsymbol{x}_k) \|$. When it is smaller than some threshold, we can obtain an upper bound of $\mathcal{O}(\eta^2)$. Otherwise, when $\| \nabla f(\boldsymbol{x}_k) \|$ is greater than the threshold, $h_k$ is of order $\Omega(1/\| \nabla f(\boldsymbol{x}_k) \|)$, then the left hand terms are all of order $\mathcal{O}(\eta^2 h_k \| f(\boldsymbol{x}) \|^2)$. Therefore we scale $\eta$ small enough to make left hand terms smaller than $\alpha \eta h_k \| f(\boldsymbol{x}) \|^2/4$. At last, we sum up the respective upper bounds together to conclude the lemma.

*Remark* C.2. Moreover, we remark that the thresholds chosen during proof ($r + \tau_0$ for proving (23) and $\tau_0/(1 - \tau_1)$ for proving (24) and (25)) are quite artificial. A thorough investigation towards these thresholds would definitely improve the dependence on the constants $(L_0, L_1, \tau_0, \tau_1)$, but would not affect our main argument.

*Proof.* In fact, this lemma can be proved in an obvious way by separating into different cases.

**(i)** If $\| \nabla f(\boldsymbol{x}_k) \| \leq r + \tau_0$, it directly follows that $L_1 \| \nabla f(\boldsymbol{x}_k) \| \eta^2 d\sigma^2/2 \leq L_1(r + \tau_0)\eta^2 d\sigma^2/2$; otherwise, if $\| \nabla f(\boldsymbol{x}_k) \| > r + \tau_0$, we know

$$
h_k = \frac{1}{r + \| \boldsymbol{g}_k \|} \geq \frac{1}{r + \tau_0 + (\tau_1 + 1)\| \nabla f(\boldsymbol{x}_k) \|} \geq \frac{1}{3 \| \nabla f(\boldsymbol{x}_k) \|},
$$

therefore $\eta \leq \alpha/(6L_1 d\sigma^2)$ directly yields

$$\frac{L_1\|\nabla f(\boldsymbol{x}_k)\|\eta^2 d\sigma^2}{2} \leq \frac{\eta\alpha h_k\|\nabla f(\boldsymbol{x}_k)\|^2}{4}.$$

Then (23) follows from summing up these two cases.

**(ii)** If $\|\nabla f(\boldsymbol{x}_k)\| \leq \tau_0/(1-\tau_1)$, then $\|\boldsymbol{g}_k - \nabla f(\boldsymbol{x}_k)\| \leq \tau_0 + \tau_1\|\nabla f(\boldsymbol{x}_k)\| \leq \tau_0/(1-\tau_1)$ and $h_k \leq 1/r$, which yield

$$(L_0 + L_1\|\nabla f(\boldsymbol{x}_k)\|)\eta^2 h_k^2 \langle \nabla f(\boldsymbol{x}_k), \boldsymbol{g}_k - \nabla f(\boldsymbol{x}_k)\rangle \leq \frac{(L_0(1-\tau_1) + L_1\tau_0)\tau_0^2}{r^2(1-\tau_1)^3}\eta^2,$$

$$\frac{L_0 + L_1\|\nabla f(\boldsymbol{x}_k)\|}{2}\eta^2 h_k^2\|\boldsymbol{g}_k - \nabla f(\boldsymbol{x}_k)\|^2 \leq \frac{(L_0(1-\tau_1) + L_1\tau_0)\tau_0^2}{2r^2(1-\tau_1)^3}\eta^2.$$

Otherwise, if $\|\nabla f(\boldsymbol{x}_k)\| > \tau_0/(1-\tau_1)$, we note that $\|\boldsymbol{g}_k\| \geq -\tau_0 + (1-\tau_1)\|\nabla f(\boldsymbol{x}_k)\|$ and

$$h_k(L_0 + L_1\|\nabla f(\boldsymbol{x}_k)\|) \leq \frac{L_0 + L_1\|\nabla f(\boldsymbol{x}_k)\|}{r - \tau_0 + (1-\tau_1)\|\nabla f(\boldsymbol{x}_k)\|} \leq \max\left(\frac{L_0}{r-\tau_0}, \frac{L_1}{1-\tau_1}\right).$$

Consequently, once $\eta \leq \dfrac{\alpha}{4}\min\left(\dfrac{r-\tau_0}{L_0}, \dfrac{1-\tau_1}{L_1}\right)$, we have

$$(L_0 + L_1\|\nabla f(\boldsymbol{x}_k)\|)\eta^2 h_k^2 \langle \nabla f(\boldsymbol{x}_k), \boldsymbol{g}_k - \nabla f(\boldsymbol{x}_k)\rangle$$
$$\leq \max\left(\frac{L_0}{r-\tau_0}, \frac{L_1}{1-\tau_1}\right)\eta^2 h_k\|\nabla f(\boldsymbol{x}_k)\|^2 \leq \frac{\eta\alpha h_k}{4}\|\nabla f(\boldsymbol{x}_k)\|^2,$$

and

$$\frac{L_0 + L_1\|\nabla f(\boldsymbol{x}_k)\|}{2}\eta^2 h_k^2\|\boldsymbol{g}_k - \nabla f(\boldsymbol{x}_k)\|^2$$
$$\leq \frac{1}{2}\max\left(\frac{L_0}{r-\tau_0}, \frac{L_1}{1-\tau_1}\right)\eta^2 h_k\|\nabla f(\boldsymbol{x}_k)\|^2 \leq \frac{\eta\alpha h_k}{4}\|\nabla f(\boldsymbol{x}_k)\|^2.$$

We obtain (24) and (25) via summing up respective bounds for two cases.

**(iii)** The last bound (26) can be derived directly by

$$\frac{L_0 + L_1\|\nabla f(\boldsymbol{x}_k)\|}{2}\eta^2 h_k^2\|\nabla f(\boldsymbol{x}_k)\|^2 \leq \max\left(\frac{L_0}{r-\tau_0}, \frac{L_1}{\tau_1}\right)\frac{\eta^2}{2}h_k\|\nabla f(\boldsymbol{x}_k)\|^2 \leq \frac{\eta\alpha h_k}{4}\|\nabla f(\boldsymbol{x}_k)\|^2$$

via setting $\eta \leq \dfrac{\alpha}{2}\min\left(\dfrac{r-\tau_0}{L_0}, \dfrac{\tau_1}{L_1}\right)$.

In conclusion, it suffices to set $\eta \leq \min\left(\dfrac{(r-\tau_0)\alpha}{4L_0}, \dfrac{(1-\tau_1)\alpha}{4L_1}, \dfrac{\alpha}{6L_1 d\sigma^2}\right)$ to obtain these bounds. $\qquad\square$

In the sequel, we will use this lemma only with

$$\alpha = \alpha_0 := \frac{\tau_0(1-\tau_1)}{2r(1-\tau_1) + 4\tau_0} < \frac{1}{4} \tag{27}$$

**Lemma C.2.** *In the statement of Theorem 3.2, we take $\eta = \sqrt{\frac{2}{L_1(r+\tau_0)Td\sigma^2}}$. Then the condition (22) in Lemma C.1 holds as long as we run the algorithm long enough i.e. $T \geq C\left(\sigma^2, \tau, L, d, r\right)$.*

*Proof.* We see that

$$\eta = \sqrt{\frac{2}{L_1(r+\tau_0)Td\sigma^2}} \leq \min\left(\frac{(r-\tau_0)\alpha_0}{4L_0}, \frac{(1-\tau_1)^2\alpha_0}{4L_1}, \frac{\alpha_0}{6L_1d\sigma^2}\right)$$

is equivalent to

$$T \geq \max\left(\frac{32L_0^2}{(r-\tau_0)^2\alpha_0^2L_1(r+\tau_0)d\sigma^2}, \frac{32L_1}{\tau_1^2\alpha_0^2(r+\tau_0)d\sigma^2}, \frac{72L_1d}{\alpha_0^2(r+\tau_0)}\right).$$

$\square$

## C.2   Final Procedures in Proof

*Proof of Theorem 3.2.* Equipped with Lemmas B.3 ,C.1 and C.3, we further wrote the one step inequality (21) into

$$\eta A(\|\nabla f(\boldsymbol{x}_k)\|)$$
$$\leq f(\boldsymbol{x}_k) - \mathbb{E}_k\left[f(\boldsymbol{x}_{k+1})\right] + \eta^2\left(\frac{(L_1(r+\tau_0)+L_0)d\sigma^2}{2} + \frac{3(L_0(1-\tau_1)+L_1\tau_0)\tau_0^2}{2r^2(1-\tau_1)^3}\eta^2\right). \quad (28)$$

We then separate the time index into

$$\mathcal{U} = \left\{k < T : \|\nabla f(\boldsymbol{x}_k)\| \geq \frac{\tau_0}{1-\tau_1}\right\}$$

and $\mathcal{U}^c = \{0, 1, \cdots, T-1\}\backslash\mathcal{U}$. Given this, we derive from (17) that for any $k \in \mathcal{U}$,

$$\frac{\tau_0}{2r(1-\tau_1)+4\tau_0}\eta\|\nabla f(\boldsymbol{x}_k)\|$$
$$\leq \mathbb{E}_k\left[f(\boldsymbol{x}_{k+1})\right] - f(\boldsymbol{x}_k) + \eta^2\left(\frac{(L_1(r+\tau_0)+L_0)d\sigma^2}{2} + \frac{3(L_0(1-\tau_1)+L_1\tau_0)\tau_0^2}{2r^2(1-\tau_1)^3}\right).$$

Similarly, together with $\alpha_0 \leq 1/4$, (18) deduces that for any $k \in \mathcal{U}^c$,

$$\eta\left[\frac{3(1-\tau_1)}{4r(1-\tau_1)+8\tau_0}\|\nabla f(\boldsymbol{x}_k)\|^2 - \frac{4\tau_0^3}{r(r+\tau_0)(1-\tau_1)^3}\right]$$
$$\leq \mathbb{E}_k\left[f(\boldsymbol{x}_{k+1})\right] - f(\boldsymbol{x}_k) + \eta^2\left(\frac{(L_1(r+\tau_0)+L_0)d\sigma^2}{2} + \frac{3(L_0(1-\tau_1)+L_1\tau_0)\tau_0^2}{2r^2(1-\tau_1)^3}\right).$$

Sum these inequalities altogether to have

$$\frac{1}{r+2\tau_0}\max\left\{\frac{\tau_0}{2T}\sum_{k\in\mathcal{U}}\|\nabla f(\boldsymbol{x}_k)\|, \frac{3(1-\tau_1)}{4T}\sum_{k\notin\mathcal{U}}\|\nabla f(\boldsymbol{x}_k)\|^2\right\}$$
$$\leq\frac{D_f}{T\eta} + \eta\frac{(L_1(r+\tau_0)+L_0)d\sigma^2}{2} + \eta\frac{3(L_0(1-\tau_1)+L_1\tau_0)\tau_0^2}{2r^2(1-\tau_1)^3} + \frac{4\tau_0^3}{r(r+\tau_0)(1-\tau_1)^3}\frac{|\mathcal{U}^c|}{T}.$$

In order to optimize the first two terms, we set

$$\eta = \sqrt{\frac{2}{(L_1(r+\tau_0)+L_0)Td\sigma^2}}. \quad (29)$$

We then define

$$\Delta = (D_f+1)\sqrt{\frac{(L_1(r+\tau_0)+L_0)d\sigma^2}{2T}} + \frac{3(L_0+L_1\tau_0)\tau_0^2}{2r^2(1-\tau_1)^3}\sqrt{\frac{2}{(L_1(r+\tau_0)+L_0)Td\sigma^2}} + \frac{4\tau_0^3}{r(r+\tau_0)(1-\tau_1)^3}.$$

Recall that $\sigma^2$ can have some dependence on $T$, so actually these three terms are $\mathcal{O}(\sigma/\sqrt{T})$, $\mathcal{O}(1/(\sqrt{T}\sigma))$ and $\mathcal{O}(1)$ respectively. Then we further have

$$
\mathbb{E}\left[\min_{0 \le k < T}\|\nabla f(\boldsymbol{x}_k)\|\right]
$$

$$
\le \mathbb{E}\left[\min\left\{\sqrt{\frac{1}{|\mathcal{U}|}\sum_{k \in \mathcal{U}^c}\|\nabla f(\boldsymbol{x}_k)\|^2}, \frac{1}{|\mathcal{U}|}\sum_{k \notin \mathcal{U}}\|\nabla f(\boldsymbol{x}_k)\|\right\}\right]
$$

$$
\le \max\left\{\sqrt{\frac{8(r+2\tau_0)}{3(1-\tau_1)}\Delta}, \frac{2(r+2\tau_0)}{\tau_0}\Delta\right\}, \tag{30}
$$

where the second inequality follows from the fact that either $|\mathcal{U}| \ge T/2$ or $|\mathcal{U}^c| \ge T/2$. In the end, we capture the leading terms in this upper bound to have

$$
\mathbb{E}\left[\min_{0 \le k < T}\|\nabla f(\boldsymbol{x}_k)\|\right] \le \mathcal{O}\left(\sqrt[4]{\frac{(D_f+1)^2(L_1(r+\tau_0)+L_0)(r+2\tau_0)^2 d\sigma^2}{T(1-\tau_1)^2}}\right)
$$

$$
+ \mathcal{O}\left(\sqrt{\sqrt{\frac{2(r+2\tau_0)^2}{(L_1(r+\tau_0)+L_0)Td\sigma^2}}\frac{3(L_0+L_1\tau_0)\tau_0^2}{2r^2(1-\tau_1)^4} + \frac{8(r+2\tau_0)\tau_0^2}{r(r+\tau_0)(1-\tau_1)^3}}\right).
$$

$\square$

**Lemma C.3.** *In the statement of Corollary 3.3, we take $\eta = \sqrt{\frac{2}{(L_1(r+\tau_0)+L_0)Td\sigma^2}}$, $\sigma = c_2\sqrt{T\log\frac{1}{\delta}}/(N\epsilon)$ and $T \ge \mathcal{O}(N^2\epsilon^2/(r^3 d\log\frac{1}{\delta}))$. Then the condition in Lemma C.2 holds as long as we have enough samples i.e. $N \ge C(\epsilon, \delta, \tau, L, B, d, r)$.*

*Proof.* It is more straight-forward to verify the condition (22) directly. Firstly, we plug $\sigma^2 = \frac{c_2^2 T\log(1/\delta)}{N^2\epsilon^2}$ from Lemma 3.1 into the formula of $\eta$ to have

$$
\eta = \sqrt{\frac{2}{(L_1(r+\tau_0)+L_0)Td\sigma^2}} = \frac{N\epsilon}{c_2 T}\sqrt{\frac{2}{(L_1(r+\tau_0)+L_0)d\log(1/\delta)}} \le \frac{\alpha_0 N^2\epsilon^2}{6L_1 dc_2^2 T\log(1/\delta)} = \frac{\alpha_0}{6L_1 d\sigma^2}
$$

as long as we have enough samples

$$
N \ge \frac{6c_2 L_1}{\epsilon\alpha_0}\sqrt{\frac{2d\log(1/\delta)}{L_1(r+\tau_0)+L_0}}.
$$

Other conditions

$$
\eta = \frac{N\epsilon}{c_2 T}\sqrt{\frac{2}{(L_1(r+\tau_0)+L_0)d\log(1/\delta)}} \le \min\left\{\frac{(r-\tau_0)\alpha_0}{4L_0}, \frac{(1-\tau_1)\alpha_0}{4L_1}\right\}
$$

holds as long as we run the algorithm long enough

$$
\frac{T}{N} \ge \min\left\{\frac{L_0}{r-\tau_0}, \frac{L_1}{\tau_1}\right\}\frac{4\epsilon}{\alpha_0 c_2}\sqrt{\frac{2}{(L_1(r+\tau_0)+L_0)d\log(1/\delta)}}. \tag{31}
$$

The last requirement (31) naturally holds due to $T \ge \mathcal{O}(N^2\epsilon^2/(r^3 d\log\frac{1}{\delta}))$. $\square$

*Proof of Corollary 3.3.* Moreover, we take the limit $T \ge \mathcal{O}(N^2\epsilon^2/(B^2 r^3 d\log\frac{1}{\delta}))$ to derive the privacy-utility trade-off

$$
\mathbb{E}\left[\min_{0 \le k < T}\|\nabla f(\boldsymbol{x}_k)\|\right] \le \mathcal{O}\left(\sqrt[4]{\frac{(D_f+1)^2(L_1(r+\tau_0)+L_0)d\log(1/\delta)}{N^2\epsilon^2}} + \frac{8(r+2\tau_0)\tau_0^3}{r(r+\tau_0)(1-\tau_1)^3}\right),
$$

ending the proof. $\square$

# D  Proofs for DP-SGD

In this section, we prove the convergence theorem for DP-SGD following the roadmap outlined in Section C. To start with, we extend (19) into

$$
\begin{aligned}
&\mathbb{E}_k\left[f(\boldsymbol{x}_{k+1})\right] - f(\boldsymbol{x}_k) \\
&\leq -\eta\mathbb{E}_k\left[\bar{h}_k\right]\|\nabla f(\boldsymbol{x}_k)\|^2 - \eta\mathbb{E}_k\left[\langle\bar{h}_k\nabla f(\boldsymbol{x}_k),\boldsymbol{g}_k - \nabla f(\boldsymbol{x}_k)\rangle\right] \\
&\quad + \frac{L_0 + L_1\|\nabla f(\boldsymbol{x}_k)\|}{2}\eta^2\left(\mathbb{E}_k\left[\bar{h}_k^2\|\nabla f(\boldsymbol{x}_k)\|^2\right] + 2\mathbb{E}_k\left[\bar{h}_k^2\langle\boldsymbol{g}_k - \nabla f(\boldsymbol{x}_k),\nabla f(\boldsymbol{x}_k)\rangle\right]\right) \\
&\quad + \frac{L_0 + L_1\|\nabla f(\boldsymbol{x}_k)\|}{2}\eta^2\left(dc^2\sigma^2 + \mathbb{E}_k\left[\bar{h}_k^2\|\boldsymbol{g}_k - \nabla f(\boldsymbol{x}_k)\|^2\right]\right).
\end{aligned}
\tag{32}
$$

## D.1  Upper Bound Second-order Terms

In the same spirit as Lemma C.1, we provide an upper bound for the second-order terms in the following lemma.

**Lemma D.1.** *For any $0 < \alpha < 1$ to be determined explicitly later, if*

$$
\eta \leq \min\left\{\frac{\alpha}{6L_1dc\sigma^2}, \frac{\alpha(1-\tau_1)}{2L_0(1-\tau_1)+4L_1\tau_0}, \frac{\alpha\tau_0(1-\tau_1)}{4c(L_0(1-\tau_1)+2L_1\tau_0)}\right\},
\tag{33}
$$

*then we have*

$$
\frac{L_0+L_1\|\nabla f(\boldsymbol{x}_k)\|}{2}\eta^2 dc^2\sigma^2 \leq \frac{L_1(c+\tau_0)+L_0}{2}\eta^2 dc^2\sigma^2 + \frac{\alpha\eta\bar{h}_k}{4}\|\nabla f(\boldsymbol{x}_k)\|^2,
\tag{34}
$$

$$
(L_0+L_1\|\nabla f(\boldsymbol{x}_k)\|)\eta^2\bar{h}_k^2\langle\nabla f(\boldsymbol{x}_k),\boldsymbol{g}_k - \nabla f(\boldsymbol{x}_k)\rangle \leq \frac{2(L_0(1-\tau_1)+2L_1\tau_0)\tau_0^2}{(1-\tau_1)^3}\eta^2 + \frac{\alpha\eta\bar{h}_k}{4}\|\nabla f(\boldsymbol{x}_k)\|^2,
\tag{35}
$$

$$
\frac{L_0+L_1\|\nabla f(\boldsymbol{x}_k)\|}{2}\eta^2\bar{h}_k^2\|\boldsymbol{g}_k - \nabla f(\boldsymbol{x}_k)\|^2 \leq \frac{2(L_0(1-\tau_1)+2L_1\tau_0)\tau_0^2}{(1-\tau_1)^3}\eta^2 + \frac{\alpha\eta\bar{h}_k}{4}\|\nabla f(\boldsymbol{x}_k)\|^2,
\tag{36}
$$

$$
\frac{L_0+L_1\|\nabla f(\boldsymbol{x}_k)\|}{2}\eta^2\bar{h}_k^2\|\nabla f(\boldsymbol{x}_k)\|^2 \leq \frac{\alpha\eta\bar{h}_k}{4}\|\nabla f(\boldsymbol{x}_k)\|^2.
\tag{37}
$$

*Proof.* In fact, this lemma can be proved in an obvious way by separating into different cases.

**(i)** If $\|\nabla f(\boldsymbol{x}_k)\| \leq c + \tau_0$, it directly follows that $L_1\|\nabla f(\boldsymbol{x}_k)\|\eta^2 d\sigma^2/2 \leq L_1(c+\tau_0)\eta^2 dc^2\sigma^2/2$; otherwise, if $\|\nabla f(\boldsymbol{x}_k)\| > c + \tau_0$, we know

$$
\bar{h}_k = \min\left\{1, \frac{c}{\|\boldsymbol{g}_k\|}\right\} \geq \frac{c}{c+\|\boldsymbol{g}_k\|} \geq \frac{c}{c+\tau_0+(\tau_1+1)\|\nabla f(\boldsymbol{x}_k)\|} \geq \frac{c}{3\|\nabla f(\boldsymbol{x}_k)\|}
$$

therefore $\eta \leq \alpha/(6L_1dc\sigma^2)$ directly yields

$$
\frac{L_1\|\nabla f(\boldsymbol{x}_k)\|\eta^2 dc^2\sigma^2}{2} \leq \frac{\eta\alpha\bar{h}_k\|\nabla f(\boldsymbol{x}_k)\|^2}{4}.
$$

Then (34) follows from summing up these two cases.

**(ii)** If $\|\nabla f(\boldsymbol{x}_k)\| \leq 2\tau_0/(1-\tau_1)$, then $\|\boldsymbol{g}_k - \nabla f(\boldsymbol{x}_k)\| \leq \tau_0 + \tau_1\|\nabla f(\boldsymbol{x}_k)\| \leq 2\tau_0/(1-\tau_1)$ and $\bar{h}_k \leq 1$, which yield

$$
(L_0+L_1\|\nabla f(\boldsymbol{x}_k)\|)\eta^2\bar{h}_k^2\langle\nabla f(\boldsymbol{x}_k),\boldsymbol{g}_k - \nabla f(\boldsymbol{x}_k)\rangle \leq \frac{2(L_0(1-\tau_1)+2L_1\tau_0)\tau_0^2}{(1-\tau_1)^3}\eta^2,
$$

$$
\frac{L_0+L_1\|\nabla f(\boldsymbol{x}_k)\|}{2}\eta^2\bar{h}_k^2\|\boldsymbol{g}_k - \nabla f(\boldsymbol{x}_k)\|^2 \leq \frac{2(L_0(1-\tau_1)+2L_1\tau_0)\tau_0^2}{(1-\tau_1)^3}\eta^2.
$$

Otherwise, if $\|\nabla f(\boldsymbol{x}_k)\| > 2\tau_0/(1 - \tau_1)$, we note that $\|\boldsymbol{g}_k - \nabla f(\boldsymbol{x}_k)\| \leq \frac{1+\tau_1}{2}\|\nabla f(\boldsymbol{x}_k)\|$ and $\|\boldsymbol{g}_k\| \geq \frac{1-\tau_1}{2}\|\nabla f(\boldsymbol{x}_k)\|$. Moreover,

$$\bar{h}_k(L_0 + L_1\|\nabla f(\boldsymbol{x}_k)\|) \leq \frac{c(L_0 + L_1\|\nabla f(\boldsymbol{x}_k)\|)}{\|\boldsymbol{g}_k\|} \leq \frac{2c(L_0 + L_1\|\nabla f(\boldsymbol{x}_k)\|)}{(1 - \tau_1)\|\nabla f(\boldsymbol{x}_k)\|} \leq \frac{cL_0}{\tau_0} + \frac{2cL_1}{1 - \tau_1}.$$

Consequently, once $\eta \leq \frac{\alpha}{4}\frac{\tau_0(1 - \tau_1)}{c(L_0(1 - \tau_1) + 2L_1\tau_0)}$, we have

$$(L_0 + L_1\|\nabla f(\boldsymbol{x}_k)\|)\eta^2\bar{h}_k^2\langle\nabla f(\boldsymbol{x}_k), \boldsymbol{g}_k - \nabla f(\boldsymbol{x}_k)\rangle$$
$$\leq \left(\frac{cL_0}{\tau_0} + \frac{2cL_1}{1 - \tau_1}\right)\eta^2\bar{h}_k\|\nabla f(\boldsymbol{x}_k)\|^2 \leq \frac{\eta\alpha\bar{h}_k}{4}\|\nabla f(\boldsymbol{x}_k)\|^2,$$

and

$$\frac{L_0 + L_1\|\nabla f(\boldsymbol{x}_k)\|}{2}\eta^2\bar{h}_k^2\|\boldsymbol{g}_k - \nabla f(\boldsymbol{x}_k)\|^2$$
$$\leq \frac{1}{2}\max\left(\frac{cL_0}{\tau_0} + \frac{2cL_1}{1 - \tau_1}\right)\eta^2\bar{h}_k\|\nabla f(\boldsymbol{x}_k)\|^2 \leq \frac{\eta\alpha\bar{h}_k}{4}\|\nabla f(\boldsymbol{x}_k)\|^2.$$

We obtain (35) and (36) via summing up respective bounds for two cases.

**(iii)** We firstly derive a bound on $\bar{h}_k(L_0 + L_1\|\nabla f(\boldsymbol{x}_k)\|)$. When $\|\nabla f(\boldsymbol{x}_k)\| \leq 2\tau_0/(1 - \tau_1)$, we know $\bar{h}_k(L_0 + L_1\|\nabla f(\boldsymbol{x}_k)\|) \leq \frac{L_0(1-\tau_1)+2L_1\tau_0}{1-\tau_1}$. Otherwise, we know $\bar{h}_k(L_0 + L_1\|\nabla f(\boldsymbol{x}_k)\|) \leq \frac{cL_0}{\tau_0} + \frac{2cL_1}{1-\tau_1}$. The last bound (37) can be derived directly by

$$\frac{L_0 + L_1\|\nabla f(\boldsymbol{x}_k)\|}{2}\eta^2\bar{h}_k^2\|\nabla f(\boldsymbol{x}_k)\|^2$$
$$\leq \max\left(\frac{L_0(1 - \tau_1) + 2L_1\tau_0}{1 - \tau_1}, \frac{cL_0}{\tau_0} + \frac{2cL_1}{1 - \tau_1}\right)\frac{\eta^2}{2}\bar{h}_k\|\nabla f(\boldsymbol{x}_k)\|^2 \leq \frac{\eta\alpha\bar{h}_k}{4}\|\nabla f(\boldsymbol{x}_k)\|^2$$

via setting $\eta \leq \frac{\alpha}{2}\min\left(\frac{1 - \tau_1}{L_0(1 - \tau_1) + 2L_1\tau_0}, \frac{\tau_0(1 - \tau_1)}{c(L_0(1 - \tau_1) + 2L_1\tau_0)}\right).$

In general, the four inequalities hold as long as we ensure (33). $\qquad\square$

Explanations in Remarks C.1 and C.2 also explain the motivations behind this proof. In the sequel, we will use this lemma only with

$$\alpha = \alpha_0 := \frac{\tau_0(1 - \tau_1)}{c(1 - \tau_1) + 2\tau_0} < \frac{1}{2}. \tag{38}$$

**Lemma D.2.** *In the statement of Theorem 3.4, we take* $\eta = \sqrt{\frac{2}{(L_1(c+\tau_0)+L_0)Tdc^2\sigma^2}}$. *Then the condition (33) in Lemma D.1 holds as long as we run the algorithm long enough i.e.* $T \geq C\left(\sigma^2, \tau, L, d, c\right)$.

*Proof.* We see that

$$\eta = \sqrt{\frac{2}{(L_1(c + \tau_0) + L_0)Tdc^2\sigma^2}} \leq \min\left\{\frac{\alpha_0}{6L_1dc\sigma^2}, \frac{\alpha_0(1 - \tau_1)}{2L_0(1 - \tau_1) + 4L_1\tau_0}, \frac{\alpha_0\tau_0(1 - \tau_1)}{4c(L_0(1 - \tau_1) + 2L_1\tau_0)}\right\}$$

is equivalent to

$$T \geq \frac{2}{(L_1(c + \tau_0) + L_0)dc^2\sigma^2}\max\left\{\frac{6L_1dc\sigma^2}{\alpha}, \frac{2L_0(1 - \tau_1) + 4L_1\tau_0}{\alpha(1 - \tau_1)}, \frac{4c(L_0(1 - \tau_1) + 2L_1\tau_0)}{\alpha\tau_0(1 - \tau_1)}\right\}^2.$$

$\qquad\square$

## D.2 Final Procedures in Proof

*Proof of Theorem 3.4.* Equipped with Lemmas B.5, D.1 and D.3, we further wrote the one step inequality (32) into

$$
\begin{aligned}
\eta B(\|\nabla f(\boldsymbol{x}_k)\|) \leq &f(\boldsymbol{x}_k) - \mathbb{E}_k\left[f(\boldsymbol{x}_{k+1})\right] \\
&+ \eta^2\left(\frac{(L_1(c+\tau_0)+L_0)dc^2\sigma^2}{2} + \frac{4(L_0(1-\tau_1)+2L_1\tau_0)\tau_0^2}{(1-\tau_1)^3}\eta^2\right).
\end{aligned}
\tag{39}
$$

We then separate the time index into

$$
\mathcal{U} = \left\{k < T : \|\nabla f(\boldsymbol{x}_k)\| \geq \frac{\tau_0}{1-\tau_1}\right\}
$$

and $\mathcal{U}^c = \{0, 1, \cdots, T-1\}\backslash\mathcal{U}$. Given this, we derive from (39) that for any $k \in \mathcal{U}$,

$$
\begin{aligned}
&\frac{\tau_0(c-1)}{c(1-\tau_1)+2\tau_0}\eta\|\nabla f(\boldsymbol{x}_k)\| \\
&\leq f(\boldsymbol{x}_k) - \mathbb{E}_k\left[f(\boldsymbol{x}_{k+1})\right] + \eta^2\left(\frac{(L_1(c+\tau_0)+L_0)dc^2\sigma^2}{2} + \frac{4(L_0(1-\tau_1)+2L_1\tau_0)\tau_0^2}{(1-\tau_1)^3}\eta^2\right).
\end{aligned}
$$

Similarly, together with $\alpha_0 \leq 1/2$, (18) deduces that for any $k \in \mathcal{U}^c$,

$$
\begin{aligned}
&\frac{1}{2}\eta\|\nabla f(\boldsymbol{x}_k)\|^2 \\
&\leq f(\boldsymbol{x}_k) - \mathbb{E}_k\left[f(\boldsymbol{x}_{k+1})\right] + \eta^2\left(\frac{(L_1(c+\tau_0)+L_0)dc^2\sigma^2}{2} + \frac{4(L_0(1-\tau_1)+2L_1\tau_0)\tau_0^2}{(1-\tau_1)^3}\eta^2\right).
\end{aligned}
$$

Sum these inequalities altogether to have

$$
\begin{aligned}
&\max\left\{\frac{\tau_0(c-1)}{c(1-\tau_1)+2\tau_0}\frac{1}{T}\sum_{k\in\mathcal{U}}\|\nabla f(\boldsymbol{x}_k)\|, \frac{1}{2T}\sum_{k\notin\mathcal{U}}\|\nabla f(\boldsymbol{x}_k)\|^2\right\} \\
&\leq \frac{(f(x_0)-f^*)}{T\eta} + \eta\frac{(L_1(c+\tau_0)+L_0)dc^2\sigma^2}{2} + \eta\frac{4(L_0(1-\tau_1)+2L_1\tau_0)\tau_0^2}{(1-\tau_1)^3}.
\end{aligned}
$$

We minimize the sum of first two terms by setting

$$
\eta = \sqrt{\frac{2}{(L_1(c+\tau_0)+L_0)Tdc^2\sigma^2}}.
\tag{40}
$$

We then define

$$
\Delta = (D_f+1)\sqrt{\frac{(L_1(c+\tau_0)+L_0)dc^2\sigma^2}{2T}} + \frac{4(L_0+2L_1\tau_0)\tau_0^2}{(1-\tau_1)^3}\sqrt{\frac{2}{(L_1(c+\tau_0)+L_0)Tdc^2\sigma^2}}.
\tag{41}
$$

Recall $\sigma^2$ can grow with $T$, so these two terms are $\mathcal{O}(\sigma/\sqrt{T}), \mathcal{O}(1/(\sigma\sqrt{T}))$ respectively. Then we further have

$$
\begin{aligned}
&\mathbb{E}\left[\min_{0\leq k<T}\|\nabla f(\boldsymbol{x}_k)\|\right] \\
&\leq \mathbb{E}\left[\min\left\{\sqrt{\frac{1}{|\mathcal{U}|}\sum_{k\in\mathcal{U}^c}\|\nabla f(\boldsymbol{x}_k)\|^2}, \frac{1}{|\mathcal{U}|}\sum_{k\notin\mathcal{U}}\|\nabla f(\boldsymbol{x}_k)\|\right\}\right] \\
&\leq \max\left\{\sqrt{4\Delta}, \frac{2(c+2\tau_0)}{\tau_0(c-1)}\Delta\right\},
\end{aligned}
\tag{42}
$$

where the second inequality follows from the fact that either $|\mathcal{U}| \geq T/2$ or $|\mathcal{U}^c| \geq T/2$. In the end, we capture the leading terms in this upper bound to have

$$
\mathbb{E}\left[\min_{0 \leq k < T} \|\nabla f(\boldsymbol{x}_k)\|\right] \leq \mathcal{O}\left(\sqrt[4]{\frac{(D_f + 1)^2 (L_1(c + \tau_0) + L_0) dc^2 \sigma^2}{2T}}\right)
$$
$$
+ \mathcal{O}\left(\sqrt{\sqrt{\frac{2}{(L_1(c + \tau_0) + L_0) T dc^2 \sigma^2}} \frac{(L_0 + 2L_1 \tau_0) \tau_0}{(1 - \tau_1)^3}}\right).
$$

$\square$

**Lemma D.3.** *In the statement of Corollary 3.5, we take* $\eta = \sqrt{\frac{2}{(L_1(c+\tau_0)+L_0) T dc^2 \sigma^2}}$, $\sigma = c_2 B\sqrt{T \log(1/\delta)}/(N\epsilon)$ *and* $T \geq \mathcal{O}(N^2\epsilon^2/(B^2 c^3 d \log \frac{1}{\delta}))$. *Then the condition in Lemma D.2 holds as long as we have enough samples i.e.* $N \geq C(\epsilon, \delta, \tau, L, B, d, c)$.

*Proof.* It is more straight-forward to verify the condition (22) directly. Firstly, we plug $\sigma^2 = \frac{c_2^2 T \log(1/\delta)}{N^2 \epsilon^2}$ from Lemma 3.1 into the formula of $\eta$ to have

$$
\eta = \sqrt{\frac{2}{(L_1(c+\tau_0)+L_0) T dc^2 \sigma^2}} = \frac{N\epsilon}{c_2 T}\sqrt{\frac{2}{(L_1(c+\tau_0)+L_0) dc^2 \log(1/\delta)}} \leq \frac{\alpha_0 N^2 \epsilon^2}{6 L_1 dc^2 c_2^2 T \log(1/\delta)} = \frac{\alpha_0}{6 L_1 dc^2 \sigma^2}
$$

as long as we have enough samples

$$
N \geq \frac{6 L_1 c c_2}{\epsilon \alpha_0}\sqrt{\frac{2d \log(1/\delta)}{L_1(c+\tau_0)+L_0}}.
$$

Other conditions

$$
\eta = \frac{N\epsilon}{cc_2 T}\sqrt{\frac{2}{(L_1(c+\tau_0)+L_0) d \log(1/\delta)}} \leq \min\left\{\frac{\alpha_0(1-\tau_1)}{2L_0(1-\tau_1)+4L_1\tau_0}, \frac{\alpha_0\tau_0(1-\tau_1)}{4c(L_0(1-\tau_1)+2L_1\tau_0)}\right\}
$$

holds as long as we run the algorithm long enough

$$
\frac{T}{N} \geq \min\left\{1, \frac{2c}{\tau_0}\right\}\frac{2(L_0(1-\tau_1)+2L_1\tau_0)\epsilon}{cc_2\alpha_0(1-\tau_1)}\sqrt{\frac{2}{(L_1(c+\tau_0)+L_0) d \log(1/\delta)}}. \tag{43}
$$

The last requirement (43) naturally holds due to $T \geq \mathcal{O}(N^2\epsilon^2/(B^2 c^3 d \log \frac{1}{\delta}))$. $\square$

*Proof of Corollary 3.5.* Moreover, we take the limit $T \geq \mathcal{O}(N^2\epsilon^2/(B^2 c^3 d \log \frac{1}{\delta}))$ to derive the privacy-utility trade-off

$$
\mathbb{E}\left[\min_{0 \leq k < T} \|\nabla f(\boldsymbol{x}_k)\|\right] \leq \mathcal{O}\left(\sqrt[4]{\frac{(D_f + 1)^2 (L_1(c + \tau_0) + L_0) dc^2 \log(1/\delta)}{N^2 \epsilon^2}}\right),
$$

ending the proof. $\square$

## E Proof for Privacy Guarantee

This section presents a simple proof for Lemma 3.1. To begin with, we formally introduce the functional view of Renyi Differential Privacy below. Define a functional as

$$
\epsilon_{\mathcal{M}}(\alpha) \triangleq \sup_{\mathbb{D}, \mathbb{D}'} D_\alpha(\mathcal{M}(\mathbb{D}) \| \mathcal{M}(\mathbb{D}')) = \sup_{\mathbb{D}, \mathbb{D}'} \frac{1}{\alpha - 1}\log \mathbb{E}_{\theta \sim \mathcal{M}(\mathbb{D}')}\left[\left(\frac{\mathcal{M}(\mathbb{D})(\theta)}{\mathcal{M}(\mathbb{D}')(\theta)}\right)^\alpha\right], \alpha \geq 1 \tag{44}
$$

where $\mathcal{M}(\mathbb{D})$ denotes the distribution of the output with input $\mathbb{D}$ and $\mathcal{M}(\mathbb{D})(\theta)$ refers to the density at $\theta$ of this distribution. The following propositions clarify several notions of differential privacy in the literature.

**Proposition E.1.** *Let $\mathcal{M}$ be a randomized mechanism.*

**(i)** *If and only if $\epsilon_{\mathcal{M}}(\infty) \leq \epsilon$, then $\mathcal{M}$ is $\epsilon$-(pure)-DP (Dwork et al., 2014a).*

**(ii)** *If and only if $\epsilon_{\mathcal{M}}(\alpha) \leq \epsilon$, then $\mathcal{M}$ is $(\alpha, \epsilon)$-RDP (Renyi differential privacy) (Mironov, 2017).*

**(iii)** *If and only if $\delta \geq \exp[(\alpha - 1)(\epsilon_{\mathcal{M}}(\alpha) - \epsilon)]$ for some $\alpha \geq 1$, then $\mathcal{M}$ is $(\epsilon, \delta)$-DP (Dwork et al., 2014b).*

**(iv)** *If and only if $\epsilon_{\mathcal{M}}(\alpha) \leq \rho\alpha$ for any $\alpha \geq 1$, then $\mathcal{M}$ is $\rho$-zCDP (zero-concentrated differential privacy) (Bun & Steinke, 2016).*

**(v)** *If and only if $\epsilon_{\mathcal{M}}(\alpha) \leq \rho\alpha$ for any $\alpha \in (1, \omega)$, then $\mathcal{M}$ is $(\rho, \omega)$-tCDP (truncated concentrated differential privacy) (Bun et al., 2018).*

We remark that Proposition E.1(iii) is adapted from the second assertion of Theorem 2 in Abadi et al. (2016), while the literature prefers to use the converse argument for this assertion, Proposition 3 in Mironov (2017). Here we also restate the composition theorem for Renyi differential privacy.

**Proposition E.2** (Proposition 1, Mironov (2017)). *Let $\mathcal{M} = \mathcal{M}_T \circ \mathcal{M}_{T-1} \circ \cdots \circ \mathcal{M}_1$ be defined in an interactively compositional way, then for any fixed $\alpha \geq 1$,*

$$\epsilon_{\mathcal{M}}(\alpha) \leq \sum_{i=1}^{T} \epsilon_{\mathcal{M}_i}(\alpha).$$

DP-SGD and DP-NSGD under our consideration can both be decomposed into $T$ composition of sub-sampled Gaussian mechanism with *uniform sampling without replacement*, denoted as $\texttt{Gaussian}(\sigma) \circ \texttt{subsample}(N, B)$. We write the privacy-accountant functional, (44), of this building-block mechanism as $\hat{\epsilon}(\alpha)$.

It is widely known that the sole Gaussian mechanism has $\epsilon_{\texttt{Gaussian}(\sigma)}(\alpha) = \alpha/(2\sigma^2)$, Table II in Mironov (2017), when the $\ell_2$-sensitivity of the unperturbed mechanism is normalized to 1. The sub-sampled Gaussian mechanism is much more complicated and draws many previous efforts. In particular, Abadi et al. (2016); Mironov et al. (2019) study *Poisson sub-sampling*, which is less popular in practical sub-sampling; Wang et al. (2019b) proposed a general bound for any *uniformly* sub-sampled RDP mechanisms, but their bound is a bit loose when restricted to Gaussian mechanisms. Thankfully, Bun et al. (2018) developed a general privacy-amplification bound for any *uniformly* sub-sampled tCDP mechanisms, which is satisfying for our later treatment. Specifically, we specify Theorem 11 in Bun et al. (2018) to the Gaussian mechanism, to get the following proposition.

**Proposition E.3** (Privacy Amplification by Uniform Sub-sampling without Replacement). *For the very mechanism $\texttt{Gaussian}(\sigma) \circ \texttt{subsample}(N, B)$ with $B < 0.1N$, we have the following privacy accountant*

$$\hat{\epsilon}(\alpha) \leq \frac{7\gamma^2\alpha}{\sigma^2}, \quad \forall \alpha \leq \frac{\sigma^2}{2}\log\left(\frac{1}{\gamma}\right), \tag{45}$$

*with $\gamma = B/N$.*

*Proof of Lemma 3.1.* We denote whole composited mechanism as $\mathcal{M}$. We view the summation of clipped/normalized gradients as the unperturbed mechanism, so the Gaussian noise we add is $\mathcal{N}(0, \sigma^2 B^2 c^2)$ for DP-SGD and $\mathcal{N}(0, \sigma^2 B^2)$. However, their respective privacy guarantee are still the same, since DP-SGD has $\ell_2$ sensitivity $c$ while DP-NSGD has $\ell_2$ sensitivity 1. By Propositions E.2 and E.3, we have

$$\epsilon_{\mathcal{M}}(\alpha) \leq \frac{7T\gamma^2\alpha}{B^2\sigma^2}, \quad \forall \alpha \leq \frac{B^2\sigma^2}{2}\log\left(\frac{1}{\gamma}\right).$$

Further by Proposition E.1(iii), DP-SGD is $(\epsilon, \delta)$-DP if there exists $\alpha \leq \frac{B^2\sigma^2}{2}\log\left(\frac{1}{\gamma}\right)$ such that

$$7T\gamma^2\alpha/(B^2\sigma^2) \leq \epsilon/2,$$
$$\exp(-(\alpha - 1)\epsilon/2) \leq \delta.$$

Plus, we find that when $\epsilon = c_1\gamma^2 T$, we can satisfy all these conditions by setting

$$\sigma \geq c_2 \frac{\gamma\sqrt{T\log(1/\delta)}}{B\epsilon}$$

for some explicit constants $c_1$ and $c_2$. $\qquad\square$

