# OpenReview forum: "Normalized/Clipped SGD with Perturbation for Differentially Private Non-Convex Optimization"
_TMLR — Rejected by TMLR_

### Review · Reviewer_LAfU · 2023-11-03

**Summary Of Contributions:**

This paper studies the convergence behavior of differentially private versions of stochastic gradient descent (SGD), in particular DP-SGD (where gradients are clipped) and DP-NSGD (where gradients are normalized).
The paper gets to bounds on the convergence rate which seem similar to those of existing work.

**Audience:**

Yes

**Broader Impact Concerns:**

No ethical concerns.

**Claims And Evidence:**

No

**Requested Changes:**

Clearly, the text needs to be made much more rigorous, all language mistakes need to be removed, and more importantly all mathematics should be made very precise.
Next, it is important to compare with existing work, and either position the paper as outperforming the state of the art or as an alternative approach which achieves similar results (in which case it is useful to highlight how the paper brings new insight).

**Strengths And Weaknesses:**

Understanding better the convergence rate of DP-SGD and DP-NSGD is an interesting goal.  It is unclear how exactly the paper improves over the state of the art.  Table 1 shows that existing algorithms achieve a similar convergence rate, but this Table 1 marks existing algorithms as not "handling bias".  As far as I understand from the algorithms described in the text, the paper doesn't propose an alternative which avoids bias in the gradients.  On page 10 I read "As the training goes, the gradient norm becomes small and clipping has no effect ..." which is not necessarily true, as not the average gradient is clipped but the individual gradients (on individual instances) are clipped.  These individual gradients can still be of significant magnitude, even if they cancel each other at the point of convergence.  It is possible that at convergence some gradients poiting in one direction are clipped much more strongly than gradients pointing in the opposite direction, making the point of convergence shift with respect to the optimal solution.  As a result, I don't see how the proposed method "handles bias", and how the proposed method is significantly better than existing approaches from the point of view of the listed bounds.

As the detailed comments below illustrate for the first few pages, the text contains a large number of minor issues (language mistakes, undefined notations, implicit assumptions ...), the text is clearly insufficiently rigorous in its current form.





Detailed comments:
* Eq (1) : due to sampling errors, the second equality is not exact.
* page 2 top: "also referred to as gradient perturbation approach" : "approach" needs an article (the, a)
* page 2 : "An intuitive thought that one may favor DP-NSGD" -< ".. that may favor DP-NSGD" ?
* page 2 : while it is true that the threshold $c$ may be hard to choose, how about choosing $r$ ?
* "If properly setting the hyperparameters, we achieve O(...) convergence rate" -> "... we achieve an O(...) convergence rate".
* Contributions, last bullet: "We evaluate their empirical performance" -> the word "their" probably refers to DP-SGD and DP-NSGD but these are not the last things mentioned, hence it is better to explicitly mention DP-SGD and DP-NSGD here.
* Table 1: it is unclear when exactly you consider that an algorithm "handles bias".
* Sec 2.1: "defined for every model parameters" -> "... parameter"
* "and sample instance $\xi$" -> "and sample instance $\xi_i$"
* "In the seuqel," -> "... sequel"
* Before the end of definition 2.1, Section 2.1 has already referred to $\xi_i$ as a sample, an instance and a data record. Please avoid such synonyms and consistently use the same term to avoid confusion.  Especially the word "sample" has other meanings (being also a verb).
* Paragraph after Def 2.1: "including drawing gradients estimates" -> "... gradient estimates".
* "$\mathbb{E}_k$ takes conditional expectation given $x_k$" -> Please first define $x_k$.  Note that Eq (2) used $x^+$, so it is not obvious here that $k$ denotes some iteration number.
* "In the non-convex setting," -> I guess you mean "In settings where the loss is not convex in $x$", but up to here the text didn't make this explicit.
* "If we want to measure utility via bounding function values, extra convex condition or its weakened versions are essential." -> The grammar of this sentence has issues, probably with missing articles are singular/plural mismatches.
* Sec 2.2: "the smoothness $\|\nabla^2 f(x)\|$" -> The notation $\nabla^2$ is commonly used to denote the Hessian rather than some measure of smoothness.
* "This relaxed notion of smoothness, Definition 2.2, and lower bounded function value together constitute our first assumption below. " -> This sentence doesn't make much sense to me.  In the assumption below, I don't see a "lower bounded function value".
* "We assume that $f(x)$ is $(L_0 , L_1 )$-generalized smooth, Definition 2.2." -> " .... smooth, as defined in Definition 2.2.".
* "We also assume $D_f = f(x_0) - f^* < \infty$" -> please define x_0.
* "We do not assume an upper bound on $D_X = \|x_0 − x^*\|$" -> please define $x^*$.
* Assumption 2: In $\|\nabla_x \ell(x,\xi) - \nabla f(x)\|$, what is the value of $\xi$ to be used?  Did you forget an expectation or an average over the dataset?

---

> ### Author Response · Authors · 2024-02-10
> **Response to Reviewer LAfU**
>
> We thank the reviewer for the detailed comments on the manuscript and we have taken careful editing on our text. We highlight the following bullet point to address the reviewer’s main concerns:
>
> **Question1: It is unclear how exactly the paper improves over the state of the art. Table 1 shows that existing algorithms achieve a similar convergence rate, but this Table 1 marks existing algorithms as not "handling bias". As far as I understand from the algorithms described in the text, the paper doesn't propose an alternative which avoids bias in the gradicents.**
>
> **Response1:** As highlighted in the draft, our aim is not to eliminate the bias introduced by gradient clipping/normalization, a common and default practice in private machine learning. Instead, our primary goal is to offer a theoretical analysis of how this bias impacts convergence, which in turn affects the utility of the model. The proposed DP-NSGD is not bias-free but exhibits comparable convergence analysis with DP-SGD. The text has been revised to clarify our motivation more clearly. We have changed Table 1 correspondingly.
>
> **Question2: On Page 10, which reads, "As training goes, the gradient norm becomes small, and clipping has no effect…”, which is not necessarily true.**
>
> **Response2:** We want to clarify that as training goes, most data points would be well-fitted, and the magnitude of their gradients becomes small. We acknowledge that there may still be data points that are not yet fitted and have large magnitude of gradients. Comparatively speaking, gradient normalization (DP-NSGD) tends to amplify the impact of samples with relatively small gradients, thereby exhibits quite different training curves from gradient clipping (DP-SGD). It is important to note that our method does not aim to eliminate bias from a methodological standpoint. Moreover, in the revised manuscript, we have changed the “good samples” to “well-fitted samples” for clarification.
>
> **Question3: Language mistakes, undefined notations, implicit assumptions.**
>
> **Response3:**
> We are grateful to the reviewer for highlighting these issues. In response, we have meticulously revised the language throughout the paper based on the reviewer's suggestions.
> Below are our selected responses to the detailed comments.
>
> 1) Equation (1) is indeed valid as it pertains to evaluating optimization performance using the empirical average as the objective. Whether constructing a mini batch involves uniform subsampling with or without replacement, both approaches align with this objective. We appreciate the reviewer's insight into the potential confusion caused by our original wording. Accordingly, we have included additional comments around that equation to improve clarity.
>
> 2) In Table 1, we have introduced an additional footnote to elaborate on the "handle bias" column. This column is designed to indicate whether the theoretical analysis addresses the effects of gradient clipping/regularization or circumvents these issues through strong assumptions about the gradients.
>
> 3) The paragraphs discussing the assumptions have been thoroughly revised for clarity and precision.

---

> > ### Comment · Reviewer_LAfU · 2024-02-10
> >
> > > Question2: On Page 10, which reads, "As training goes, the gradient norm becomes small, and clipping has no effect…”, which is not necessarily true.
> >
> > > Response2: We want to clarify that as training goes, most data points would be well-fitted, and the magnitude of their gradients becomes small. We acknowledge that there may still be data points that are not yet fitted and have large magnitude of gradients. Comparatively speaking, gradient normalization (DP-NSGD) tends to amplify the impact of samples with relatively small gradients, thereby exhibits quite different training curves from gradient clipping (DP-SGD). It is important to note that our method does not aim to eliminate bias from a methodological standpoint. Moreover, we have changed the “good samples” to “well-fitted samples” for clarification.
> >
> > It seems a common misunderstanding that "the sum of gradients is zero" implies "all gradients are small".  In practice, at least for gradient clipping (classic DP-SGD), unless the clipping threshold is chosen very high (causing also a very large privacy cost), experiments on at least some common datasets (MNIST, CIFAR etc) suggest that the sum-of-gradients becomes zero and the size of individual gradients decreases a bit (the fit of individual instances improves a bit) but not much (i.e., the majority of instances whose gradient needed clipping in the beginning also need clipping after convergence).  It is possible the situation is different for DP-NSGD, but at least I would like to see some evidence before you make strong claims such as "clipping has no effect".  Else, saying "has a somewhat smaller effect" would be safer.
> >
> > Unlike what your response seems to suggest, not only DP-NSGD but also DP-SGD tend to amplify the importance of instances with relatively smaller gradients (or more correctly, bound the impact of instances with large gradients).

---

> ### Author Response · Authors · 2024-02-11
> **Thanks for the further clarification**
>
> Dear Reviewer LAfU,
>
> We thank you for the further clarification and appreciate your detailed feedback. Regarding your concern on the percentage of clipping in DP-SGD, a recent study by He et al. (2023), titled "Exploring the Limits of Differentially Private Deep Learning with Group-Wise Clipping," uses an adaptive quantile estimation method for setting the clipping threshold. This method aims for a target quantile of 0.85, implying that 85% of gradients remain unclipped at each iteration. Their experiments focus on large language model fine-tuning tasks and have observed that during the latter stages of differentially private training, sample gradients exhibit two distinct patterns: well-fitted data gradients become very small, while not-yet-fitted data gradients (around 15% of the total samples, which varies from task to task) remain significantly larger. However, it's important to note that these observations may not directly transfer to differentially private image classification scenarios. We would like to take the reviewer's suggestion and make the claim safer.
>
> Specifically, we will refine our statement to more accurately reflect the comparative impact of DP-NSGD and DP-SGD on the amplification of the importance of instances with smaller gradients as follows: "While both DP-NSGD and DP-SGD tend to amplify the significance of instances with small gradients, the amplification effect of DP-NSGD is generally more pronounced than that of DP-SGD." This adjustment aims to provide a more nuanced understanding of the effects of gradient clipping in the context of differentially private optimization techniques.

---

### Review · Reviewer_xft1 · 2023-12-15

**Summary Of Contributions:**

The authors study two variations of DP-SGD based on either clipping or normalizing the per-sample gradients. The convergence rate is studied under weaker assumptions on the objective function than in the literature, specifically $(L_0, L_1)$-smoothness and $(\tau_0, \tau_1)$-bounded gradient variance. A key observation is that the performance of the normalization version depends heavily on the regularization factor used.

**Audience:**

Yes

**Broader Impact Concerns:**

None.

**Claims And Evidence:**

No

**Requested Changes:**

- An in-depth comparison with Bu et al. (2022), preferably both theoretical and empirical.
- Improve clarity based on the suggestions outlined above.
- Update the bibliography.

**Strengths And Weaknesses:**

DP-SGD is arguably the most widely used approach to obtain privacy guarantees in large-scale machine learning (deep learning in particular). Hence, any improvement of DP-SGD is likely to be of interest to TMLR’s audience.

Overall, I think the ideas presented are quite natural and reasonably well-motivated, though I haven’t checked the proofs in the Appendix and can’t make a statement regarding their correctness.

**Clarity of presentation**
On the positive side, I appreciate the discussion after Corollary 3.3 and the pedagogical examples that the authors bring up. The clarity of presentation could, however, be improved in other places, e.g.
- Variables are sometimes used without being defined, e.g. $\xi$ and $\mathcal{S}$ in the introduction.
- Additional refs would be appropriate in several places. For instance, in the introduction, I think it would’ve been appropriate to cite: DP-SGD, a few examples of applications, and the claim that it is widely observed in deep learning that the gradient variance grows with the norm of the gradient. Furthermore, in section 4.1, you say “As in literature, we replace…” which definitely merits a reference.
- There are quite a few grammatical errors/typos (e.g. "Apart from the literature mentioned in Introduction, there are a large body of works", "seuqel") and occasionally unclear formulations like “our utility bounds are best available under even weakened conditions” and “we adopt uniform sub-sampling without sampling” (do you mean without replacement?).
- Maybe I missed something, but I didn’t understand how you determine appropriate values of constants such as $L_0$ and $L_1$ for complex functions such as the neural networks used in the experiments.

**Evidence**
My impression is that the work is a resubmission of an older work that hasn’t undergone major revision. This view is based on two observations: (1) the authors refer to Bu et al. (2022) as “concurrent work”, which is definitely a stretch, and (2) the most recent papers cited are from around mid-2022. I don’t have a problem with resubmissions per se, but I do think the paper would need to be updated to reflect the *current* state of the art.

In particular, the biggest weakness in my view is that the authors do not provide a thorough comparison with the work of Bu et al. (2022) who also study the convergence of DP-NSGD. Instead, it is dismissed based on the underlying assumptions being “unrealistic” and “artificial”. I think a detailed theoretical and/or empirical investigation of the differences between the authors’ work and the analysis of Bu et al. (2022) is warranted.

Related to the above comments, I note that the bibliography is partially outdated, e.g.
- Bu et al. (2022) was published in NeurIPS 2023.
- “Improved Convergence of Differential Private SGD with Gradient Clipping” was published in ICLR 2023.
- “Efficient privacy-preserving stochastic nonconvex optimization” was published in UAI 2023.

---

> ### Author Response · Authors · 2024-02-10
> **Response to Reviewer xft1**
>
> We thank the reviewer for the careful reading and valuable suggestions on our manuscript. The following are our responses.
>
>
> **Question1: Clarity of presentation could be improved in other places**.
>
> **Response1:**  We have revised the paper according to the suggestions. Here is a list of changes.
>
> 1) the definitions of $\xi$ and $\mathcal{S}$ are added at the first place where they appear in the Introduction.
>
> 2) More references are added based on the reviewers’ suggestions.
>
> 3) We have corrected the typos in the manuscript and have made a round of proofreading.
>
> 4) The hyper-parameters in our empirical experiments are tuned by grid search. The values of L_0 and L_1 do not have to be determined before running the algorithm, which is not directly reflected in the algorithm either.
>
> **Question2:  A thorough comparison with the work of Bu et al. (2022) who also study the convergence of DP-NSGD.**
>
> **Response2:** We have made several revisions in the manuscript to do the comparison, which are highlighted in blue. 1) We add a comparison in the convergence summary table (Table 1). 2) We add a paragraph in the end of Section 3.3 to theoretically compare the results between Bu et al. (2023) and ours. 3) We add a sentence to acknowledge the extensive empirical results that are done in Bu et al. (2023) in the end of experiment section.
>
> **Qestion3: Update the bibliography.**
>
> **Response3:** We have updated the bibliography accordingly. Thank the reviewer for the kind suggestion.

---

### Review · Reviewer_4pJb · 2024-01-27

**Summary Of Contributions:**

This paper proposes and analyzes two algorithms, DP-SGD and DP-NSGD, for training deep learning models with differential privacy guarantees. Both algorithms clip or normalize the per-sample gradients to limit sensitivity before adding noise for privacy. The paper provides convergence analysis for nonconvex objectives, showing both algorithms can achieve a rate of O(d/(N^2ε^2)) for the gradient norm under mild assumptions. Empirically, DP-NSGD and DP-SGD achieve comparable accuracy on vision and language tasks, but DP-NSGD is easier to tune. The analysis introduces a regularizer for DP-NSGD that controls the bias-noise tradeoff.

**Audience:**

Yes

**Claims And Evidence:**

Yes

**Requested Changes:**

Paper should discuss about convergence rates obtained for non-convex function from DP Riemannian Optimization literature [1, 2, 3]  in related work.  Note some non-convex functions are geodesically convex. Hence with Riemannian framework utility bounds are measured in the expected empirical excess risk $E[F(w_{priv})]− F(w)]$ for geodesically convex function not just $E[||\nabla  F(w_{priv})||_2^2 ]$, which traditional optimization tools cannot provide.





Refs


[1] Differentially private Riemannian optimization. arXiv preprint arXiv:2205.09494 (2022).

[2] Improved Differentially Private Riemannian Optimization: Fast Sampling and Variance Reduction. Transactions on Machine Learning Research (2022).

[3] Differentially Private Fréchet Mean on the Manifold of Symmetric Positive Definite (SPD) Matrices with log-Euclidean Metric. Transactions on Machine Learning Research (2022)

**Strengths And Weaknesses:**

Strengths:

The paper provides strong theoretical convergence guarantees for DP-SGD and DP-NSGD under mild assumptions, improving over prior work. The analysis carefully handles the bias induced by gradient clipping/normalization.
Empirically, DP-NSGD and DP-SGD achieve comparable accuracy on vision and language tasks. DP-NSGD seems easier to tune than DP-SGD, which could help save privacy budget during hyperparameter tuning.

Weaknesses:

The assumptions, while weaker than prior work, are still quite strong compared to real-world deep learning scenarios. Analyzing convergence under more realistic assumptions would strengthen the theory. More rigorous empirical evaluation on a diverse set of tasks could better demonstrate the practical utility of the algorithms.

---

> ### Author Response · Authors · 2024-02-10
> **Response to Reviewer 4pJb**
>
> We thank the reviewer for the careful reading and insightful feedback. We have already added the literature on differentially private Riemannian optimization in the revised version.
>
> We also thank the reviewer for the comments and suggestions regarding the assumptions made in our work and the scope of our empirical evaluation. We appreciate the opportunity to address these concerns and provide additional context.
>
> Firstly, we acknowledge the critique regarding the strength of our assumptions compared to real-world deep learning scenarios. However, it is important to emphasize that our assumptions have been justified to closely (much closer than the usual L-smoothness assumption) align with actual deep learning situations, see Zhang et al. 2020b. At the same time, real-world scenarios are complex and diverse, making it challenging to craft theoretical assumptions that accurately depict every scenario. Developing an insightful and accurate assumption is one of the most valuable contributions of theoretical work beyond the scope of this work.
>
> Regarding the empirical evaluation of our algorithms, our intention was to demonstrate the properties of these optimizers and their practical utility. The selection of tasks for our evaluation was designed to showcase how these optimizers perform under various conditions, rather than provide an exhaustive validation across all conceivable tasks. It is noteworthy that the optimizers we discuss have already achieved commendable results in practice. For further evidence of their practical effectiveness, we encourage a review of concurrent work Bu et al. 2022 and subsequent studies that have utilized these optimizers in diverse real-world applications Li et al. 2022 (Large Language Models Can Be Strong Differentially Private Learners).
>
> We believe that our work strikes a balance between theoretical rigor and practical relevance, contributing valuable insights in differentially private deep learning. Taking into account your feedback improves the presentation of the draft.

---

### Decision · Action_Editor_GtvA · 2024-04-07

**Recommendation:** Reject

**Comment:**

Apologies for the slow response - with conflicting reviews and unresponsive reviewers, I had to read the paper in detail, which together with trying to wait for responses from reviewers took some time.

As said, the recommendations from the reviewers are divided. 4pJb and xft1 recommend acceptance, although xft1 is hesitant. LAfU, on the other hand, feels that their comments were not sufficiently addressed by the authors.

In order to address LAfU's concerns and TMLR review criteria, I would recommend the authors to carefully review all the claims made in the paper and make sure sufficient evidence is provided.

One concrete place to check is Sec. 5, where you claim

> We have achieved a rate that significantly improves over previous literature under similar setup and have analyzed the bias induced by the clipping or normalizing operation.

Looking at Table 1, it is not clear to me how your rate is significantly better than previous research.

Similarly, in the abstract you claim that your rate

> [...] improves over previous bounds under much weaker assumptions.

Again, it is not clear how this improves over previous bounds.

Based on my own reading of the paper, I have some additional comments the authors should address:

1. Proposition E.1(iii) claims that all $(\epsilon, \delta)$-DP mechanisms are RDP as well. This is clearly wrong, because there exist $(\epsilon, \delta)$-DP mechanisms that are not DP for any $\delta' < \delta$ for any $\epsilon$, while RDP implies $\delta \to 0$ as $\epsilon \to \infty$.

2. The statements of Proposition E.1(iv) and (v) need to be clarified regarding the ambiguous "for any" - does that mean "for all" or "there exists"?

3. Corollary 3.5 suggests that the gradient norm could be made arbitrarily small by decreasing the clipping threshold $c$. Can you please explain how this is not in contradiction with the known lower bounds for DP learning? Can you please also explain why in your experimental results, larger $c$ leads to better performance?

4. I would appreciate some discussion on the effect of the smoothness parameters $(L_0, L_1)$ to the convergence bounds.

5. Please explain your use of the RDP accountant in the experiments in more detail. The references you cite do not provide theory for RDP accounting for subsampling without replacement with substitute adjacency that you are using in your algorithms.

6. In caption of Fig. 3: $\delta$ should probably not be $\exp(-5)$ as currently indicated.

**Audience:**

In the initial reviews, all reviewers indicated that at least some individuals would be interested, even though the novel contribution seems rather limited. Still, I believe the paper would meet this criterion.

**Claims And Evidence:**

The paper contains some factual errors and questionable claims, so it does not meet this criterion. For more details, see "Comment" section below.

**Resubmission Of Major Revision:**

The authors may consider submitting a major revision at a later time.